# PARETOFLOW: GUIDED FLOWS IN MULTI-OBJECTIVE OPTIMIZATION

**Ye Yuan[1, 2]\*, Can (Sam) Chen[1,2]\*[†], Christopher Pal[2, 3, 4][‡] Xue Liu[1, 2][‡]**

[1]McGill, [2]MILA - Quebec AI Institute, [3]Polytechnique Montreal, [4]Canada CIFAR AI Chair

## ABSTRACT

In offline multi-objective optimization (MOO), we leverage an offline dataset of designs and their associated labels to simultaneously minimize multiple objectives. This setting more closely mirrors complex real-world problems compared to single-objective optimization. Recent works mainly employ evolutionary algorithms and Bayesian optimization, with limited attention given to the generative modeling capabilities inherent in such data. In this study, we explore generative modeling in offline MOO through flow matching, noted for its effectiveness and efficiency. We introduce *ParetoFlow*, specifically designed to guide flow sampling to approximate the Pareto front. Traditional predictor (classifier) guidance is inadequate for this purpose because it models only a single objective. In response, we propose a *multi-objective predictor guidance* module that assigns each sample a weight vector, representing a weighted distribution across multiple objective predictions. A local filtering scheme is introduced to address non-convex Pareto fronts. These weights uniformly cover the entire objective space, effectively directing sample generation towards the Pareto front. Since distributions with similar weights tend to generate similar samples, we introduce a *neighboring evolution* module to foster knowledge sharing among neighboring distributions. This module generates offspring from these distributions, and selects the most promising one for the next iteration. Our method achieves state-of-the-art performance across various tasks. Our code will be available here.

## 1 INTRODUCTION

Offline optimization Kim et al. (2025), a fundamental challenge in science and engineering, involves minimizing a black-box function using solely an offline dataset, with diverse applications ranging from molecule design Sarkisyan et al. (2016); Angermüller et al. (2020); Hu et al. (2025a) to neural architecture design Lu et al. (2023). Previous research primarily focuses on single-objective optimization, aiming to optimize a single desired property Trabucco et al. (2022); however, this fails to capture the complexities of real-world challenges that often require balancing multiple conflicting objectives, such as designing a neural architecture that demands both high accuracy and minimal parameter count Lu et al. (2023). In this study, we explore offline multi-objective optimization (MOO), leveraging an offline dataset of designs and their associated labels to simultaneously minimize multiple objectives.

The pioneering work Xue et al. (2024) adapts evolutionary algorithms Deb et al. (2002); Zhang & Li (2007) and Bayesian optimization Daulton et al. (2023); Zhang & Golovin (2020); Qing et al. (2023) to handle the offline MOO setting. Besides, some studies design controllable generative models that manage multiple properties Wang et al. (2024). However, these studies generally either focus on different settings, such as online optimization Gruver et al. (2023) and white-box optimization Yao et al. (2024), or utilize less advanced generative models, such as VAEs Wang et al. (2022). None of these studies fully exploits the potential of advanced generative modeling in offline MOO.

---

\*Equal tech contribution with random order: Can designs algorithm/drafts paper; Ye conducts experiments.
[†]Tech lead: `chencan421@gmail.com` or `can.chen@mila.quebec`.
[‡]Equal senior contribution with random order.

To bridge this gap, we employ a flow matching framework, renowned for its effectiveness and efficiency over diffusion models Lipman et al. (2023); Le et al. (2023); Polyak et al. (2024), to investigate generative modeling in offline MOO. We introduce the *ParetoFlow* method, specifically designed to guide flow sampling to approximate the Pareto front. The Pareto front is defined as the set of optimal objective values that are not dominated by any other points. As illustrated in Figure 1(a), the solid blue curve represents the Pareto front in a two-dimensional objective space for the conflicting objectives $f_1$ and $f_2$.

Traditional predictor (classifier) guidance [1] Dhariwal & Nichol (2021), focusing solely on a single objective, fails to adequately explore the entire Pareto front. As demonstrated in Figure 1(a), directing sample generation from pure noise (circles) towards a single objective, such as $f_1$ or $f_2$, yields isolated Pareto samples (pentagrams) without capturing the full spectrum of optimal samples. To address this, we propose **Module 1**, termed *multi-objective predictor guidance*, which assigns each sample a weighted distribution. This distribution, characterized by a weight vector across multiple objective predictions, guides sample generation towards its corresponding Pareto-optimal point. To navigate non-convex Pareto fronts, this module adopts a local filtering scheme, to filter out samples whose objective prediction vector deviates from the weight vector. These weight vectors uniformly cover the entire objective space, thereby effectively guiding sample generation towards the Pareto front. As shown in Figure 1(b), uniform weight vectors $\omega^{1-5}$ represent five weighted distributions over $f_1$ and $f_2$, ensuring that the generated samples (pentagrams) approximate the Pareto front.

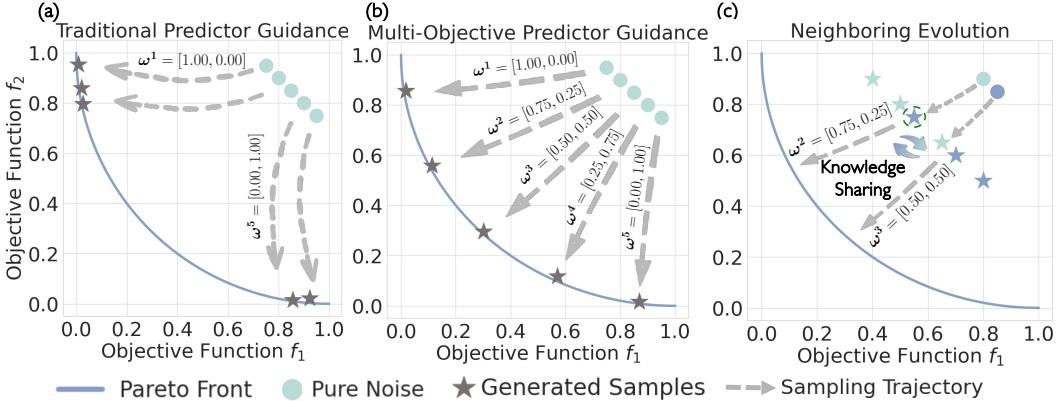

Figure 1: Motivation of **Module 1** in (b) and **Module 2** in (c).

Distributions with similar weight vectors tend to generate similar samples. As shown in Figure 1(c), the distributions with weights $\omega^2$ and $\omega^3$ are neighboring and they generate similar sample along the sampling trajectory. This motivates us to introduce **Module 2**, termed *neighboring evolution*, to foster knowledge sharing among these neighboring distributions. We propose to generate diverse offspring samples from neighboring distributions, and select the most promising one for the next iteration. For instance, consider $\omega^2$ with $\omega^3$ as its sole neighbor in Figure 1(c). We generate offspring samples from both $\omega^2$ and $\omega^3$, and then select the most promising one—identified by a dashed circle—as the next iteration state for the $\omega^2$ distribution. A similar scheme applies to $\omega^3$, assuming $\omega^2$ as its neighbor. This module facilitates valuable knowledge sharing between neighboring distributions ($\omega^2$ and $\omega^3$), enhancing the effectiveness of the overall sampling process.

To summarize, our contributions are three-fold:

- We explore the use of generative modeling in offline MOO by introducing *ParetoFlow*, specifically designed to effectively steer flow sampling to approximate the Pareto front.

- We propose a *multi-objective predictor guidance* module that assigns a uniform weighted distribution to each sample, ensuring comprehensive coverage of the objective space.

- We establish a *neighboring evolution* module to enhance knowledge sharing among distributions with close weight vectors, which improves the sampling effectiveness.

---

[1] *Classifier guidance*, initially for classification, is adapted to *predictor guidance* to generalize to regression.

## 2 PRELIMINARIES

### 2.1 OFFLINE MULTI-OBJECTIVE OPTIMIZATION

Offline multi-objective optimization (MOO) seeks to simultaneously minimize multiple objectives using an offline dataset $\mathcal{D}$ of designs and their corresponding labels. Consider a design space denoted as $\mathcal{X} \subseteq \mathbb{R}^d$, where $d$ represents the dimension of the design. In MOO, we aim to find solutions that achieve the best trade-offs among conflicting objectives. Formally, the multi-objective optimization problem is defined as:

$$\text{Find } \boldsymbol{x}^* \in \mathcal{X} \text{ such that there is no } \boldsymbol{x} \in \mathcal{X} \text{ with } \boldsymbol{f}(\boldsymbol{x}) \prec \boldsymbol{f}(\boldsymbol{x}^*), \tag{1}$$

where $\boldsymbol{f} : \mathcal{X} \rightarrow \mathbb{R}^m$ is a vector of $m$ objective functions, and $\prec$ denotes Pareto dominance. A solution $\boldsymbol{x}$ is said to *Pareto dominate* another solution $\boldsymbol{x}^*$ (denoted as $\boldsymbol{f}(\boldsymbol{x}) \prec \boldsymbol{f}(\boldsymbol{x}^*)$) if:

$$\forall i \in \{1, \dots, m\}, \quad f_i(\boldsymbol{x}) \leq f_i(\boldsymbol{x}^*) \quad \text{and} \quad \exists j \in \{1, \dots, m\} \text{ such that } f_j(\boldsymbol{x}) < f_j(\boldsymbol{x}^*). \tag{2}$$

In other words, $\boldsymbol{x}$ is no worse than $\boldsymbol{x}^*$ in all objectives and strictly better in at least one objective. A solution $\boldsymbol{x}^*$ is *Pareto optimal* if there is no other solution $\boldsymbol{x} \in \mathcal{X}$ that Pareto dominates $\boldsymbol{x}^*$. The set of all Pareto optimal solutions constitutes the *Pareto set (PS)*. The corresponding set of objective vectors, defined as $\{\boldsymbol{f}(\boldsymbol{x}) \mid \boldsymbol{x} \in PS\}$, is known as the *Pareto front (PF)*.

The goal of MOO is to identify a set of solutions that effectively approximates the PF, providing a comprehensive representation of the best possible trade-offs among the objectives.

### 2.2 FLOW MATCHING

Flow matching, an advanced generative modeling framework, excels in effectiveness and efficiency over diffusion models Lipman et al. (2023); Le et al. (2023); Polyak et al. (2024). At the core of this framework lies a conditional probability path $p_t(\boldsymbol{x} \mid \boldsymbol{x}_1), t \in [0, 1]$, evolving from an initial distribution $p_0(\boldsymbol{x} \mid \boldsymbol{x}_1) = q(\boldsymbol{x})$ to an approximate Dirac delta function $p_1(\boldsymbol{x} \mid \boldsymbol{x}_1) \approx \delta(\boldsymbol{x} - \boldsymbol{x}_1)$. This evolution is conditioned on a specific point $\boldsymbol{x}_1$ from the distribution $p_{\text{data}}$ and is driven by the conditional vector field $u_t(\boldsymbol{x} \mid \boldsymbol{x}_1)$. A neural network, parameterized by $\boldsymbol{\theta}$, learns the marginal vector field $v(\boldsymbol{x}, t)$:

$$\hat{v}(\boldsymbol{x}, t; \theta) \approx v(\boldsymbol{x}, t) = \mathbb{E}_{\boldsymbol{x}_1 \sim p_t(\boldsymbol{x}_1 | \boldsymbol{x})}[u_t(\boldsymbol{x} \mid \boldsymbol{x}_1)] \tag{3}$$

This modeled vector field, $\hat{v}(\boldsymbol{x}, t; \theta)$, functions as a neural Ordinary Differential Equation (ODE), guiding the transition from $q(\boldsymbol{x})$ to $p_{\text{data}}(\boldsymbol{x})$.

Following (Pooladian et al., 2023), the process begins by drawing initial noise $\boldsymbol{x_0}$ from $q(\boldsymbol{x_0})$. This noise is then linearly interpolated with the data point $\boldsymbol{x_1}$:

$$\boldsymbol{x} \mid \boldsymbol{x}_1, t = (1 - t) \cdot \boldsymbol{x_0} + t \cdot \boldsymbol{x_1}, \quad \boldsymbol{x_0} \sim q(\boldsymbol{x_0}) \tag{4}$$

The derivation of the conditional vector field is straightforward: $u_t(\boldsymbol{x} \mid \boldsymbol{x}_1) = (\boldsymbol{x}_1 - \boldsymbol{x})/(1 - t)$. Alternatively, this can be expressed as $u_t(\boldsymbol{x} \mid \boldsymbol{x}_1) = \boldsymbol{x}_1 - \boldsymbol{x}_0$. Training this conditional flow matching model involves optimizing the following loss function:

$$\mathbb{E}_{t, p_{\text{data}}(\boldsymbol{x}_1), q(\boldsymbol{x}_0)} \|\hat{v}(\boldsymbol{x}, t; \theta) - (\boldsymbol{x}_1 - \boldsymbol{x}_0)\|^2 \tag{5}$$

We can then use the learned vector field $\hat{v}(\boldsymbol{x}, t; \theta)$ to generate samples by solving the neural ODE.

## 3 METHOD

We describe two modules of our *ParetoFlow* method: *multi-objective predictor guidance* in Section 3.1 and *neighboring evolution* in Section 3.2. The full algorithm is detailed in Algorithm 1.

### 3.1 MULTI-OBJECTIVE PREDICTOR GUIDANCE

In this section, we first elucidate the concept of predictor guidance within the flow matching framework. Next, we detail the formulation of a weighted distribution driven by a uniform weight vector. Finally, we introduce a local filtering scheme designed to effectively manage non-convex PFs.

---

**Algorithm 1 ParetoFlow: Guided Flows in Multi-Objective Optimization**

---

**Input:** Offline dataset $\mathcal{D}$, time step $\Delta t$, number of offspring $O$, number of neighbors $K$

1: Train objective predictors $\{\hat{f}_i(\boldsymbol{x}_1; \boldsymbol{\beta}_i)\}_{i=1}^m$ for $m$ properties on $\mathcal{D}$ in a supervised manner.
2: Train the vector field $\hat{v}(\boldsymbol{x}_t, t; \boldsymbol{\theta})$ using the flow matching loss from Eq. (5).
3: Generate uniform weight vectors $\{\boldsymbol{\omega}^i\}_{i=1}^N$ using the Das-Dennis method.
4: Identify neighboring distributions for each $\boldsymbol{\omega}^i$ using Eq.(11).
5: Initialize the Pareto-optimal set $PS$ to retain high-quality samples.
6: Generate $N$ initial noise $\{\boldsymbol{x}_0^i\}_{i=1}^N$ from a standard Gaussian distribution.
7: **for** $t = 0$ **to** $1$ **do**
8:     /\*Example for a single distribution i\*/
9:     /\*This process is parallelized across N distributions\*/
10:    Set the next iteration as $s = t + \Delta t$.
11:    /\*Multi-Objective Predictor Guidance\*/
12:    Calculate the weighted distribution using Eq. (8).
13:    Compute the guided vector field $\tilde{v}(\boldsymbol{x}_t^i, t, y; \boldsymbol{\theta})$ from Eq. (9).
14:    Derive diverse samples for the $i_{th}$ distribution using Eq. (10).
15:    /\*Neighboring Evolution\*/
16:    Form the neighboring offspring set $\boldsymbol{X}_i$ based on $\mathcal{N}(i)$.
17:    Apply the local filtering scheme to filter $\boldsymbol{X}_i$ to $\boldsymbol{X}_i^l$.
18:    Select the next-iteration state $\boldsymbol{x}_s^i$ with the weighted objective using Eq. (12).
19:    If $\boldsymbol{x}_s^i$ is superior, update $PS$ with $\hat{\boldsymbol{x}}_1(\boldsymbol{x}_s^i)$.
20: **end for**
21: Return $PS$

---

**Predictor Guidance.** Originally, classifier guidance was proposed to direct sample generation toward specific image categories Dhariwal & Nichol (2021). This concept has been adapted for regression settings to guide molecule generation Lee et al. (2023); Jian et al. (2024); Chen et al. (2025). In this paper, we term this technique *predictor guidance* for a generalization. Based on *Lemma 1* in Zheng et al. (2023), we derive *predictor guidance in flow matching* as:

$$\tilde{v}(\boldsymbol{x}_t, t, y; \boldsymbol{\theta}) = \hat{v}(\boldsymbol{x}_t, t; \boldsymbol{\theta}) + \frac{1-t}{t} \nabla_{\boldsymbol{x}_t} \log p_{\boldsymbol{\beta}}(y \mid \boldsymbol{x}_t, t). \tag{6}$$

where $p_{\boldsymbol{\beta}}(y \mid \boldsymbol{x}_t, t)$ represents the predicted property distribution. Further details can be found in Appendix A.1. Training the proxy at different time steps $t$ is resource-intensive. Therefore, we approximate this by leveraging the relationship between $\boldsymbol{x}_1$ and $\boldsymbol{x}_t$:

$$p_{\boldsymbol{\beta}}(y \mid \boldsymbol{x}_t, t) = p_{\boldsymbol{\beta}}(y \mid \hat{\boldsymbol{x}}_1(\boldsymbol{x}_t), 1),$$

simplified to $p_{\boldsymbol{\beta}}(y \mid \hat{\boldsymbol{x}}_1(\boldsymbol{x}_t))$. This guides the generation of $\boldsymbol{x}_t$ towards samples with the property $y$.

**Weighted Distribution.** The preceding discussion typically pertains to generating samples to satisfy a single property $y$, whereas our framework is designed to optimize multiple properties simultaneously, denoted as $\boldsymbol{y} = [f_1(\boldsymbol{x}), \cdots, f_m(\boldsymbol{x})]$. To manage this complexity, we decompose the multi-objective generation challenge into individual weighted objective generation subproblems. Specifically, we define a weight vector $\boldsymbol{\omega} = [\omega_1, \omega_2, \cdots, \omega_m]$, where each $\omega_i > 0$ and $\sum_{i=1}^m \omega_i = 1$. The weighted property prediction is expressed as:

$$\hat{f}_{\boldsymbol{\omega}}(\boldsymbol{x}_t; \boldsymbol{\beta}) = \sum_{i=1}^m -\hat{f}_i(\hat{\boldsymbol{x}}_1(\boldsymbol{x}_t); \boldsymbol{\beta}_i)\omega_i, \tag{7}$$

where $\hat{f}_i$ predicts the $i^{th}$ objective for $\boldsymbol{x}_t$, trained using only $\boldsymbol{x}_1$ data, and the negative sign indicates minimization. We then formulate the weighted distribution as Lee et al. (2023):

$$p_{\boldsymbol{\beta}}(y \mid \hat{\boldsymbol{x}}_1(\boldsymbol{x}_t), \boldsymbol{\omega}) = e^{\gamma \hat{f}_{\boldsymbol{\omega}}(\boldsymbol{x}_t; \boldsymbol{\beta})}/Z, \tag{8}$$

where $\gamma$ is a scaling factor and $Z$ is the normalization constant. Integrating this into Eq.(6) leads to:

$$\tilde{v}(\boldsymbol{x}_t, t, y; \boldsymbol{\theta}) = \hat{v}(\boldsymbol{x}_t, t; \boldsymbol{\theta}) + \gamma \frac{1-t}{t} \nabla_{\boldsymbol{x}_t} \hat{f}_{\boldsymbol{\omega}}(\boldsymbol{x}_t; \boldsymbol{\beta}). \tag{9}$$

This vector field drives sampling towards desired properties within the weighted distribution. Equations (8) and (9) are applied in Algorithm 1, Lines 12 and 13, respectively.

Using the Das-Dennis approach Das & Dennis (1998), which subdivides the objective space into equal partitions to generate uniform weight vectors, we produce $N$ weights $\boldsymbol{\omega}$. Each weight maps to a sample, effectively covering the entire objective space. For the $i_{th}$ sample $\boldsymbol{x}_t^i$ at time step $t$, the Euler–Maruyama method Kloeden et al. (1992) is applied to advance to the next time step $\Delta t$:

$$\hat{\boldsymbol{x}}_s^i = \boldsymbol{x}_t^i + \tilde{v}(\boldsymbol{x}_t^i, t, y; \boldsymbol{\theta})\Delta t + g\sqrt{\Delta t}\epsilon, \tag{10}$$

where $s = t + \Delta t$ indicates the next time step, $g = 0.1$ denotes the noise factor, and $\epsilon$ is a standard Gaussian noise term. This process is on Line 14 in Algorithm 1. Unlike standard ODE sampling, this additional noise term $g$ enhances diversity and improves exploration of the design space.

**Local Filtering.** Using Eq. (10), sampling can reach any point on the Pareto Front (PF) if it is convex. As shown in Figure 2(a), a weight vector $\boldsymbol{\omega} = [0.5, 0.5]$ successfully guides sample generation to the $f_1 = f_2$ Pareto-optimal point. In such convex case, a set of uniform weight vectors can effectively direct sample generation across the entire PF. However, with a non-convex PF as depicted in Figure 2(b), the same weight vector skews the sampling toward favoring a single objective, either $f_1$ or $f_2$, making it challenging to approach the $f_1 = f_2$ Pareto-optimal point or its vicinity.

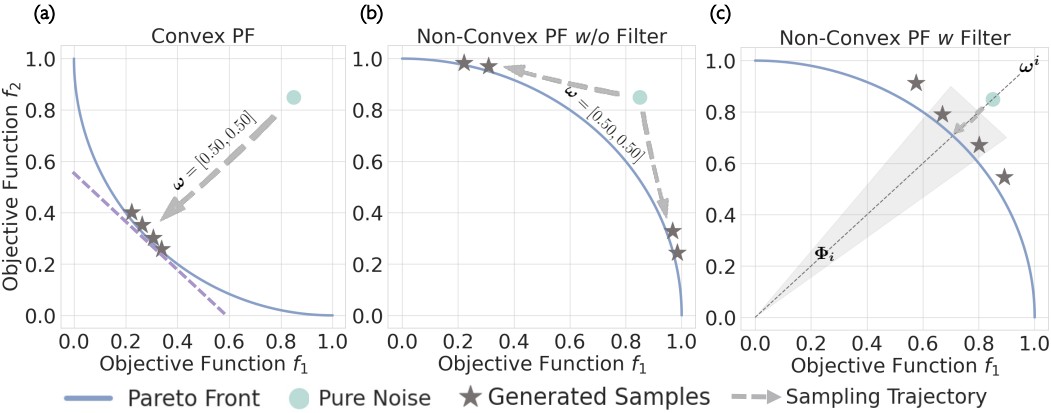

Figure 2: Local filtering: samples outside the hypercone are filtered out as shown in (c).

To overcome this, we confine the sampling space for each weighted distribution to a hypercone, characterized by an apex angle $\Phi_i$, as depicted in Figure 2(c). For any given sample $\hat{\boldsymbol{x}}_s^i$ from Eq. (10), the angle $\alpha_i$ between the prediction vector $\hat{\boldsymbol{y}}^i(\hat{\boldsymbol{x}}_s^i) = [\hat{f}_1(\hat{\boldsymbol{x}}_1(\hat{\boldsymbol{x}}_s^i); \boldsymbol{\beta}_1), \cdots, \hat{f}_m(\hat{\boldsymbol{x}}_1(\hat{\boldsymbol{x}}_s^i); \boldsymbol{\beta}_m)]$ and the weight vector $\boldsymbol{\omega}^i$ is calculated. Samples where $\alpha_i$ exceeds $\Phi_i/2$ are filtered out in Figure 2(c). Inspired by Wang et al. (2016), $\Phi_i$ is calculated as $2\sum_{j=1}^m \phi_{ij}/m$, where $\phi_{ij}$ is the angle from the $j_{th}$ closest weight to $\boldsymbol{\omega}^i$. This setup ensures that the sampled objective vectors align closely with the weight vector, enabling effective discovery of Pareto-optimal solutions at the hypercone boundaries and enhancing the diversity of the generated samples.

This local filtering scheme is also employed in the **Neighboring Update** outlined in Section 3.2 and specifically applied at Line 17 of Algorithm 1.

## 3.2 NEIGHBORING EVOLUTION

In the prior module, we discuss sampling from a single weighted distribution while overlooking the potential interactions between different distributions. In this section, we define neighboring distributions and introduce a module to foster knowledge sharing among them.

**Neighboring Distribution.** Weighted distributions with similar weight vectors are likely to produce similar samples, which could benefit from potential knowledge sharing. Since each weighted distribution is defined by a unique weight vector, we define neighboring distributions based on the proximity of their weight vectors. For a distribution associated with $\boldsymbol{\omega}^i$, its neighbors are identified as the $K$ distributions whose weight vectors have the smallest angular distances to $\boldsymbol{\omega}^i$:

$$\mathcal{N}(i) = \left\{ j : \boldsymbol{\omega}^j \in \mathrm{KNN}(\boldsymbol{\omega}^i, K, \{\boldsymbol{\omega}^l\}_{l=1}^N) \right\} \tag{11}$$

Here, $\text{KNN}(\boldsymbol{\omega}^i, K, \{\boldsymbol{\omega}^l\}_{l=1}^N)$ denotes the set of the $K$ nearest weight vectors to $\boldsymbol{\omega}_i$. By definition, distribution $i$ is also considered a neighbor of itself. It is outlined on Line 4 in Algorithm 1.

**Neighboring Update.** As mentioned, neighboring distributions generate similar samples, and we aim to leverage this similarity to foster knowledge sharing. Since $\epsilon$ introduces randomness in Eq.(10), we can obtain different next step states $\hat{\boldsymbol{x}}_s^i$ where each state can be viewed as an offspring. We can generate $O$ offspring for $\hat{\boldsymbol{x}}_s^i$, denoted as $\{\hat{\boldsymbol{x}}_s^{i,o}\}_{o=1}^O$. Given that there are $K$ neighboring samples for sample $i$, this results in a set of $K \cdot O$ offspring $\boldsymbol{X}_i = \{\hat{\boldsymbol{x}}_s^{j,o} \mid j \in \mathcal{N}(i), o \in \{1, 2, \cdots, O\}\}$. Line 16 in Algorithm 1 outlines this step. All candidates in this set are likely to satisfy the weighted distribution $i$ well, as they are guided by similar weighted distributions $j \in \mathcal{N}(i)$.

We aim to update the current sample $\boldsymbol{x}_t^i$ using the neighboring set $\boldsymbol{X}_i$. The local filtering scheme from the previous module filters out $\boldsymbol{X}_i$ to exclude objective predictions not aligned with $\boldsymbol{\omega}^i$. The remaining viable offspring are termed $\boldsymbol{X}_i^l$. Subsequently, the next iteration for $\boldsymbol{x}_t^i$ is updated as:

$$\boldsymbol{x}_s^i = \arg \max_{\hat{\boldsymbol{x}}_s^{j,o} \in \boldsymbol{X}_i^l} f_{\boldsymbol{\omega}^i}(\hat{\boldsymbol{x}}_s^{j,o}; \boldsymbol{\beta}). \tag{12}$$

This determines the next state $\boldsymbol{x}_s^i$ for each of $N$ samples and is detailed in Line 18 of Algorithm 1.

**Pareto-optimal Set Update.** While directly selecting the final $N$ samples from the flow generation is effective, we also aim to retain all high-quality samples during generation. To achieve this, we maintain a $PS$ consisting of $N$ samples, where the $i$-th sample is the best for $\hat{f}_{\boldsymbol{\omega}^i}(\cdot; \boldsymbol{\beta})$. The $PS$ is initialized with non-dominated samples in the offline dataset following Xue et al. (2024). Using Eq. (12), we compare $\boldsymbol{x}_s^i$ with the $i$-th sample in $PS$. If $\boldsymbol{x}_s^i$ is superior, we update $PS$ with $\hat{\boldsymbol{x}}_1(\boldsymbol{x}_s^i)$; otherwise, we retain the existing sample. This step is specified at Line 19 in Algorithm 1. Finally, we apply non-dominant sorting to $PS$ and select 256 candidates for evaluation.

## 4 EXPERIMENTS

We conduct comprehensive experiments to evaluate our method. In Section 4.4, we compare our approach to several baselines to assess performance. In Section 4.5, we demonstrate the effectiveness of our proposed modules.

### 4.1 BENCHMARK OVERVIEW

We utilize the Off-MOO-Bench, which summarizes and collects several established benchmarks Xue et al. (2024). We explore five groups of tasks, each task with a dataset $\mathcal{D}$ and a ground-truth oracle $\boldsymbol{f}$ for evaluation, which is not queried during training. For discrete inputs, we convert them to continuous logits as suggested by Trabucco et al. (2022); Xue et al. (2024).

**Tasks. (1)** Synthetic Function (Synthetic) Xue et al. (2024): This task encompasses several subtasks involving popular functions with 2-3 objectives, aiming to identify $PS$ with 60,000 offline designs. We exclude the DTLZ2-6 tasks as recommended by the authors due to evaluation errors [2].

**(2)** Multi-Objective Neural Architecture Search (MO-NAS) Dong & Yang (2020); Lu et al. (2023); Li et al. (2021): This task consists of multiple sub-tasks, searching for a neural architecture that optimizes multiple metrics, such as prediction error, parameter count, and GPU latency.

**(3)** Multi-Objective Reinforcement Learning (MORL) Todorov et al. (2012): **(a)** The MO-Swimmer sub-task involves finding a dimension-9,734 control policy for a robot to maximize speed and energy efficiency; **(b)** The MO-Hopper sub-task aims to find a dimension-10,184 control policy for a robot to optimize two objectives related to running and jumping.

**(4)** Scientific Design (Sci-Design): **(a)** This Molecule Zhao et al. (2021) sub-task aims to optimize two activities against biological targets GSK3$\beta$ and JNK3 in a dimension-32 molecule latent space, using 49,001 offline points. **(b)** The Regex sub-task aims to optimize protein sequences to maximize the counts of three bigrams, using 42,048 offline points. **(c)** The ZINC sub-task aims to maximize the logP (the octanol-water partition coefficient) and QED (quantitative estimate of drug-likeness) of a small molecule. **(d)** The RFP sub-task aims to maximize the solvent-accessible surface area and the stability of RFP in protein sequence designs.

---

[2]https://github.com/lamda-bbo/offline-moo/issues/14

**(5)** Real-World Applications (RE) Tanabe & Ishibuchi (2020): This category encompasses a variety of practical optimization challenges, including four-bar truss and pressure vessel design. The MO-Portfolio task Fabozzi et al. (2008) is also included here, which focuses on optimizing expected returns and variance of returns in a 20-dimensional portfolio allocation space.

The original Off-MOO-Bench also includes some combinatorial optimization tasks such as MO-TSP, MO-CVRP, and MO-KP. While these could potentially be incorporated under a generative modeling framework Sun & Yang (2023), the decoding strategy required is rather complex. As this paper focuses on a general guided flow matching method, we have opted to exclude these tasks, given the sufficient variety of other tasks already available for our evaluation.

**Evaluation.** We follow the evaluation protocol in Xue et al. (2024). Each algorithm outputs 256 solutions for evaluation. Each task has a reference point, and we compute the hypervolume metric, which measures the volume between the proposed solutions and the reference point. A larger hypervolume indicates better solutions. We report the $P$ percentile measure, employing $P = 100$ and $50$ in this study. Specifically, we rank the solutions using nondominated sorting Deb et al. (2002), remove the top $1 - P\%$ of solutions, and then report the hypervolume of the remaining solutions.

## 4.2 BASELINE METHODS

Following Xue et al. (2024), we compare two primary groups of methods: DNN-based and GP-based methods, along with some notable generative modeling methods.

**DNN-Based Methods:** These methods utilize surrogate DNN models combined with evolutionary algorithms to optimize solutions. We assess three configurations: (1) End-to-End Model (E2E): Outputs an $m$-dimensional objective vector for a design $\boldsymbol{x}$, enhanced by multi-task training techniques such as GradNorm Chen et al. (2018) and PcGrad Yu et al. (2020). (2) Multi-Head Model (MH): Uses multi-task learning to train a single predictor, employing the same techniques as End-to-End. (3) Multiple Models (MM): Maintains $m$ independent predictors, each using techniques like COMs Trabucco et al. (2021), ROMA Yu et al. (2021), IOM Qi et al. (2022), ICT Yuan et al. (2023), and Tri-mentoring Chen et al. (2023a). The default evolutionary algorithm is NSGA-II Deb et al. (2002), with results cited from the original study. Additionally, we compare MOEA/D Zhang & Li (2007) + MM due to its superior performance. We further expand our comparison to include more traditional approaches in Appendix A.2.

**GP-Based Methods:** Bayesian Optimization compute the acquisition function to select new designs, which are then evaluated using a predictor model. Techniques include: Hypervolume-based qNEHVI Daulton et al. (2021), Scalarization-based qParEGO Daulton et al. (2020), and Information-theoretic-based JES Hvarfner et al. (2022). We reference results from Xue et al. (2024).

We communicate with the authors and use the updated benchmark data for all MO-NAS tasks and the real-world application tasks RE21, RE34, RE35, RE36, RE41, RE42, and RE61. For these tasks, we rely on the latest results provided by the authors, rather than those published in the paper.

**Generative Modeling Methods:** (1) PROUD Yao et al. (2024) enhances diversity by incorporating hand-designed penalties into diffusion sampling process. (2) LaMBO-2 Gruver et al. (2023) utilizes the acquisition function to guide diffusion sample generation. (3) CorrVAE Wang et al. (2022) employs a VAE to decipher semantics and property correlations, adjusting weights in the latent space. (4) MOGFNs Jain et al. (2023) incorporates multiple objectives into the GFlowNet framework.

## 4.3 TRAINING DETAILS

Our objective is to derive 256 design samples. However, since the Das-Dennis method may not generate exactly 256 uniform weights, we generate slightly more, resulting in over 256 samples. We then use learned predictors for non-dominant sorting to select the top 256 samples. We set the number of neighboring distributions, $K$, to be $m + 1$, where $m$ is the number of objective functions, and set the number of offspring, $O$, to be 5. The sensitivity of these hyperparameters is further examined in Appendix A.3. We follow the predictor training configurations outlined in Xue et al. (2024) and flow matching training protocols described in Tomczak (2022). Additional hyperparameter details are provided in Appendix A.4 and the computational overhead is discussed in Appendix A.5.

Table 1: Average rank of different methods on each type of task in Off-MOO-Bench.

| Methods | Synthetic | MO-NAS | MORL | Sci-Design | RE | All Tasks |
|---|---|---|---|---|---|---|
| D-Best | $16.82 \pm 6.28$ | $14.42 \pm 4.11$ | $15.00 \pm 4.00$ | $13.75 \pm 6.91$ | $18.06 \pm 3.93$ | $16.02 \pm 5.13$ |
| E2E | $10.91 \pm 8.20$ | $6.05 \pm 3.32$ | $12.50 \pm 1.50$ | $9.75 \pm 4.97$ | $9.69 \pm 5.65$ | $8.73 \pm 5.88$ |
| E2E + GradNorm | $12.64 \pm 6.68$ | $13.42 \pm 5.54$ | $8.50 \pm 0.50$ | $13.50 \pm 5.12$ | $14.19 \pm 5.87$ | $13.31 \pm 5.87$ |
| E2E + PcGrad | $9.45 \pm 6.37$ | $6.42 \pm 3.18$ | $16.50 \pm 2.50$ | $14.00 \pm 3.16$ | $10.88 \pm 6.17$ | $9.40 \pm 5.70$ |
| MH | $11.55 \pm 7.19$ | $\underline{5.26 \pm 3.93}$ | $12.00 \pm 4.00$ | $12.50 \pm 3.28$ | $10.00 \pm 5.67$ | $8.87 \pm 6.00$ |
| MH + GradNorm | $10.45 \pm 6.21$ | $16.42 \pm 4.84$ | $18.00 \pm 2.00$ | $14.75 \pm 4.44$ | $17.00 \pm 4.72$ | $15.27 \pm 5.64$ |
| MH + PcGrad | $11.45 \pm 4.58$ | $6.84 \pm 2.83$ | $18.50 \pm 0.50$ | $13.50 \pm 5.41$ | $11.06 \pm 6.24$ | $10.08 \pm 5.46$ |
| MM | $\underline{4.91 \pm 4.17}$ | $6.74 \pm 3.81$ | $16.50 \pm 1.50$ | $6.75 \pm 4.32$ | $6.69 \pm 3.46$ | $\underline{6.71 \pm 4.31}$ |
| MM + COMs | $13.00 \pm 3.86$ | $9.53 \pm 4.42$ | $12.50 \pm 2.50$ | $12.25 \pm 6.83$ | $14.62 \pm 4.75$ | $12.15 \pm 5.06$ |
| MM + RoMA | $13.27 \pm 7.53$ | $8.21 \pm 5.75$ | $10.00 \pm 3.00$ | $12.00 \pm 2.45$ | $10.25 \pm 5.14$ | $10.27 \pm 6.06$ |
| MM + IOM | $6.91 \pm 3.78$ | $5.37 \pm 3.60$ | $6.50 \pm 0.50$ | $10.75 \pm 1.92$ | $7.25 \pm 4.02$ | $6.73 \pm 3.88$ |
| MM + ICT | $14.45 \pm 5.77$ | $8.53 \pm 3.12$ | $9.50 \pm 3.50$ | $12.50 \pm 7.12$ | $11.75 \pm 6.54$ | $11.12 \pm 5.77$ |
| MM + Tri-Mentor | $11.00 \pm 5.89$ | $9.05 \pm 5.71$ | $10.50 \pm 1.50$ | $13.00 \pm 3.54$ | $10.50 \pm 5.82$ | $10.27 \pm 5.65$ |
| MOEA/D + MM | $10.55 \pm 4.83$ | $12.58 \pm 5.02$ | $11.00 \pm 1.00$ | $10.75 \pm 6.87$ | $12.12 \pm 6.62$ | $11.81 \pm 5.66$ |
| MOBO | $10.91 \pm 4.42$ | $14.74 \pm 3.82$ | $17.00 \pm 0.00$ | $8.25 \pm 6.61$ | $11.00 \pm 5.79$ | $12.37 \pm 5.32$ |
| MOBO-$q$ParEGO | $13.36 \pm 3.98$ | $16.63 \pm 3.77$ | $21.00 \pm 0.00$ | $12.75 \pm 8.04$ | $17.69 \pm 4.55$ | $16.13 \pm 4.91$ |
| MOBO-JES | $17.27 \pm 3.11$ | $22.00 \pm 0.00$ | $21.00 \pm 0.00$ | $18.75 \pm 5.63$ | $13.62 \pm 5.19$ | $18.13 \pm 5.00$ |
| PROUD | $8.55 \pm 6.33$ | $14.53 \pm 4.43$ | $\underline{2.50 \pm 0.50}$ | $6.25 \pm 3.49$ | $5.75 \pm 5.02$ | $9.46 \pm 6.39$ |
| LaMBO-2 | $10.18 \pm 6.55$ | $14.37 \pm 4.66$ | $3.00 \pm 1.00$ | $\underline{5.00 \pm 1.22}$ | $\underline{5.00 \pm 4.72}$ | $9.44 \pm 6.49$ |
| CorrVAE | $11.73 \pm 6.14$ | $17.74 \pm 2.95$ | $4.50 \pm 0.50$ | $8.00 \pm 4.18$ | $9.56 \pm 6.00$ | $12.69 \pm 6.35$ |
| MOGFN | $10.55 \pm 6.04$ | $15.95 \pm 3.98$ | $3.50 \pm 1.50$ | $5.50 \pm 4.50$ | $5.88 \pm 4.97$ | $10.42 \pm 6.63$ |
| ParetoFlow (**ours**) | $\mathbf{4.00 \pm 3.88}$ | $\mathbf{3.47 \pm 4.26}$ | $\mathbf{1.00 \pm 0.00}$ | $\mathbf{2.75 \pm 1.48}$ | $\mathbf{2.44 \pm 3.45}$ | $\mathbf{3.12 \pm 3.77}$ |

## 4.4 Results and Analysis

Table 1 displays the average ranks of the 100th percentile results for all methods across various task types. Detailed hypervolume results for both the 100th and 50th percentiles are reported in Appendix A.6 and Appendix A.7, respectively. Two separator lines distinguish: (1) DNN-based methods from GP-based methods, and (2) GP-based methods from generative modeling methods. $\mathcal{D}(\text{best})$ denotes the best solution set in the offline set, characterized by the largest HV value. The last column summarizes the average rank of each method across all tasks. In each task, the best and second-best ranks are highlighted in **bold** and underlined, respectively. We provide visualization results for C-10/MOP1 and MO-Hopper, and a case study on C-10/MOP5, in Appendix A.8.

We make the following observations: **(1)** As shown in Table 1 and Figure 3, our method *ParetoFlow* consistently achieves the highest ranks across all tasks, underscoring its effectiveness. **(2)** Both DNN-based and generative modeling-based methods frequently outperform $\mathcal{D}(\text{best})$, illustrating the strength of predictor and generative modeling. **(3)** GP-based methods often underperform $\mathcal{D}(\text{best})$. We hypothesize this is because these methods, typically used in online optimization to select subsequent samples, are less effective in this offline context. **(4)** Within the generative modeling category, *ParetoFlow* surpasses other methods, including diffusion-based methods like PROUD and LaMBO-2, the VAE-based method CorrVAE, and the GFlowNet-based method MOGFN, highlighting the superiority of our *ParetoFlow* method. **(5)** MO-NAS and Sci-Design tasks are predominantly discrete, with MO-NAS having a higher dimensionality. Generative modeling methods show reduced effectiveness on MO-NAS relative to Sci-Design. This performance gap may stem from the difficulty in modeling high-dimensional discrete data.

## 4.5 Ablation Studies

Table 2: Ablation Study on ParetoFlow.

| Methods | ZDT2 | C-10/MOP1 | MO-Hopper | Zinc | RE23 |
|---|---|---|---|---|---|
| *Equal* | $6.15 \pm 0.23$ | $4.64 \pm 0.03$ | $5.58 \pm 0.38$ | $4.14 \pm 0.14$ | $4.75 \pm 0.00$ |
| *First* | $5.58 \pm 0.38$ | $4.59 \pm 0.02$ | $5.25 \pm 0.23$ | $4.00 \pm 0.14$ | $4.89 \pm 0.01$ |
| *w/o local* | $5.78 \pm 0.15$ | $4.65 \pm 0.03$ | $5.44 \pm 0.21$ | $4.36 \pm 0.04$ | $5.13 \pm 0.41$ |
| *w/o neighbor* | $6.43 \pm 0.01$ | $4.64 \pm 0.00$ | $5.62 \pm 0.16$ | $4.40 \pm 0.05$ | $6.08 \pm 0.20$ |
| *w/o PS* | $6.45 \pm 0.52$ | $4.49 \pm 0.00$ | $5.00 \pm 0.02$ | $4.40 \pm 0.00$ | $5.28 \pm 0.21$ |
| ParetoFlow | $\mathbf{6.79 \pm 0.16}$ | $\mathbf{4.77 \pm 0.00}$ | $\mathbf{5.69 \pm 0.03}$ | $\mathbf{4.49 \pm 0.06}$ | $\mathbf{6.32 \pm 0.46}$ |

We use *ParetoFlow* as the baseline to assess the impact of removing specific modules, with results detailed in Table 2. We conduct these ablation studies on representative subtasks: ZDT2 for Synthetic, C-10/MOP1 for MO-NAS, MO-Hopper for MORL, Zinc for Sci-Design, and RE23 for RE.

**Multi-Objective Predictor Guidance:** This module employs uniform weights for batch samples. In our ablation study, we explore: (1) *Equal*: Equal weight assigned to every sample across all objectives. (2) *First*: Weight applied solely to the first objective. Both variants underperform compared to the full *ParetoFlow*, demonstrating the advantage of our uniform weight scheme. *Equal* generally outperforms *First*, suggesting that focusing on a single objective can bias sample generation.

Additionally, we evaluate the impact of excluding the local filtering scheme (*w/o local*) to determine its importance. The performance drop observed without this scheme underscores its effectiveness in managing non-convex Pareto Fronts. Additionally, we measure pairwise diversity using $\frac{1}{N(N-1)} \sum_{i=1}^{N} \sum_{j=i+1}^{N} d(\boldsymbol{y}_i, \boldsymbol{y}_j)$, where $d$ denotes the Euclidean distance. This metric is applied to samples from both *ParetoFlow* and *ParetoFlow w/o local*. For *ParetoFlow w/o local*, diversity decreases from $5.144$ to $2.080$ in ZDT2, from $0.832$ to $0.827$ in C-10/MOP1, from $0.897$ to $0.181$ in MO-Hopper, from $0.721$ to $0.495$ in Zinc, and from $0.991$ to $0.814$ in RE23. This indicate that the local filtering scheme enhances performance by improving the diversity of the solution set. We further compare local filtering performance on convex and non-convex tasks in Appendix A.9. We also include in Appendix A.10 detailed comparisons between flow matching and diffusion models, as well as between the Das-Dennis method and another weight generation strategy.

**Neighboring Evolution:** We omit this module (*w/o neighbor*) to observe the effects on sample generation, focusing exclusively on direct offspring without leveraging neighboring samples. Removing this module leads to performance decrease as detailed in Table 2, demonstrating the effectiveness of neighboring information. Besides, we found that employing the neighboring module significantly improves the selection of the next step's offspring. In the sampling process, the majority of offspring selected from the neighborhood outperform those from their own distribution: $67.33\%$ for ZDT2, $73.67\%$ for C-10/MOP1, $58.33\%$ for MO-Hopper, $81.33\%$ for Zinc, and $61.98\%$ for RE23, highlighting the pivotal role of this module in the sampling process. Besides, we observe that only $12\%$ of the points in C-10/MOP1 and $1\%$ in MO-Hopper are duplicates. The higher duplication rate in C-10/MOP1 is primarily due to the decoding of continuous logits back to the same discrete values. This observation underscores the effectiveness of *ParetoFlow*.

Lastly, we examine the performance of our method without the Pareto Set ($PS$) update, relying only on the final samples produced through the sampling process. The observed performance degradation confirms the critical role of the PS update, indicating that final samples alone are insufficient.

## 5 RELATED WORK

**Offline Multi-Objective Optimization.** The primary focus of MOO research is the online setting, which involves interactive queries to a black-box function for optimizing multiple objectives simultaneously Jiang et al. (2023); Park et al. (2023); Gruver et al. (2023). However, offline MOO presents a more realistic setting, as online querying can be costly or risky Xue et al. (2024); Kim et al. (2025). In this context, two traditional methods are adapted with a trained predictor as the oracle: Evolutionary algorithms employ a population-based search strategy that includes iterative parent selection, reproduction, and survivor selection Deb et al. (2002); Zhang & Li (2007). Alternatively, Bayesian optimization leverages the learned predictor model to identify promising candidates through an acquisition function, with sampled queries advancing each iteration Daulton et al. (2023); Zhang & Golovin (2020); Qing et al. (2023). Additionally, several predictor training techniques such as COMs Trabucco et al. (2021), ROMA Yu et al. (2021), NEMO Fu & Levine (2021), ICT Yuan et al. (2023), Tri-Mentoring Chen et al. (2023a), GradNorm Chen et al. (2018), and PcGrad Yu et al. (2020) are adopted to enhance training efficacy.

Our *ParetoFlow* method is inspired by the seminal evolutionary algorithms MOEA/D Zhang & Li (2007) and LWS Wang et al. (2016), which use a weighted sum approach Ma et al. (2020) to guide populations and facilitate mutation among neighbors. The generation concept in these algorithms corresponds to the time step concept in our method. The primary distinction of our method is its generative modeling aspect: we train an advanced flow matching model on the entire dataset, enabling the exploration of data generative properties. This capability allows our sampling process to access the sample space that traditional evolutionary algorithms are unlikely to reach. We further explore

the relationship between evolutionary algorithms and flow models in our ParetoFlow framework in Appendix A.12.

**Guided Generative Modeling.** Several studies have developed generative models to produce samples meeting multiple desired properties. For instance: Wang et al. (2021) integrates structure-property relations into a conditional transformer for a biased generative process. Wang et al. (2022) employs a VAE model to recover semantics and property correlations, modeling weights in the latent space. Tagasovska et al. (2022) applies multiple gradient descent on trained EBMs to generate new samples, although training EBMs for each property can be complex. Han et al. (2023) explores a distinct setting aimed at generating modules that fulfill specific conditions. Zhu et al. (2023) uses GFlowNet as the acquisition function and Jain et al. (2023) integrates multiple objectives into GFlowNet. Yao et al. (2024) introduces diversity through hand-designed diversity penalties instead of uniform weight vectors, focusing on a white-box setting. Gruver et al. (2023) investigates online multi-objective optimization within a diffusion framework, using the acquisition to guide sample generation. Kong et al. (2024) applies multi-objective guidance under a diffusion framework but only uses equal weights for all properties, failing to capture the Pareto Front. Chen et al. (2024); Yuan et al. (2024) also explore guided diffusion models; however, their focus is limited to single-objective optimization. Instead of focusing on optimization techniques, Hu et al. (2025b) investigates auto-regressive diffusion models for molecule design, using the protein pocket as a directional condition. These studies vary in setting and approach, often using generative models that are either less advanced or challenging to train. Unlike these efforts, our work combines the advanced generative model of flow matching with evolutionary priors in traditional algorithms, an intersection never explored in the existing literature.

## 6 CONCLUSION

In this work, we apply flow matching to offline multi-objective optimization, introducing *ParetoFlow*. Our *multi-objective predictor guidance* module employs a uniform weight vector for each sample generation, guiding samples to approximate the Pareto-front. Additionally, our *neighboring evolution* module enhance knowledge sharing between neighboring distributions. Extensive experiments across various benchmarks confirm the effectiveness of our approach. We discuss ethics statement and limitations in Appendix A.13.

## 7 ACKNOWLEDGEMENTS

This research was partially funded by the Fonds de recherche du Québec – Nature et technologies. We also gratefully acknowledge CIFAR for its support through the AI Chairs program.

We thank Mattie Tesfaldet and Alexander Tong from Mila, along with Chin-Wei Huang from Microsoft Research, for their insightful discussions on score-based models. We further appreciate Jiarui Lu from Mila for his valuable suggestions regarding the presentation of this paper.

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

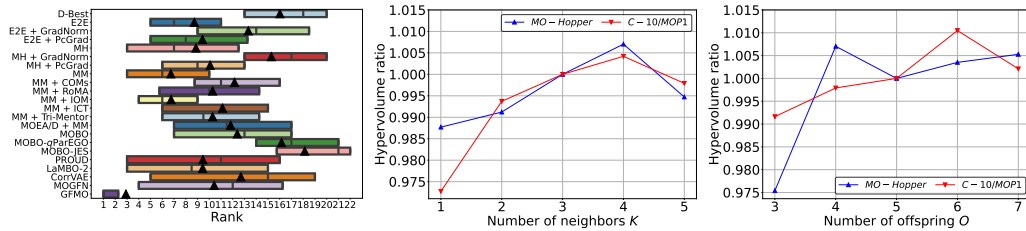

Figure 3: Sticks and triangles are rank medians and means.

Figure 4: Sensitivity to the number of neighbors $K$.

Figure 5: Sensitivity to the number of offspring $O$.

# A APPENDIX

## A.1 PREDICTOR GUIDANCE IN FLOW MATCHING

From *Lemma 1* in Zheng et al. (2023), the guided vector field is derived as:

$$\tilde{v}(\boldsymbol{x}_t, t, y; \boldsymbol{\theta}) = a_t \boldsymbol{x}_t + b_t \nabla_{\boldsymbol{x}_t} \log p(\boldsymbol{x}_t \mid y) \tag{13}$$

where $a_t = \frac{\dot{\alpha}_t}{\alpha_t}$ and $b_t = (\dot{\alpha}_t \sigma_t - \alpha_t \dot{\sigma}_t) \frac{\sigma_t}{\alpha_t}$. With $\alpha_t = t$ and $\sigma_t = 1 - t$, we simplify Eq. (13):

$$\tilde{v}(\boldsymbol{x}_t, t, y; \boldsymbol{\theta}) = \frac{1}{t} \boldsymbol{x}_t + \frac{1-t}{t} \nabla_{\boldsymbol{x}_t} \log p(\boldsymbol{x}_t \mid y) \tag{14}$$

The log-probability function is expressed as:

$$\log p(\boldsymbol{x}_t \mid y) = \log p_{\boldsymbol{\theta}}(\boldsymbol{x}_t) + \log p_{\boldsymbol{\beta}}(y \mid \boldsymbol{x}_t, t) - \log p(y) \tag{15}$$

where $p_{\boldsymbol{\theta}}(\boldsymbol{x}_t)$ represents the data distribution learned by the flow matching model and $p_{\boldsymbol{\beta}}(y \mid \boldsymbol{x}_t, t)$ denotes the predicted property distribution.

Substituting these expressions leads to:

$$\begin{aligned} \tilde{v}(\boldsymbol{x}_t, t, y; \boldsymbol{\theta}) &= \frac{1}{t} \boldsymbol{x}_t + \frac{1-t}{t} \nabla_{\boldsymbol{x}_t} \log p_{\boldsymbol{\theta}}(\boldsymbol{x}_t) + \frac{1-t}{t} \nabla_{\boldsymbol{x}_t} \log p_{\boldsymbol{\beta}}(y \mid \boldsymbol{x}_t, t) \\ &= \tilde{v}(\boldsymbol{x}_t, t; \boldsymbol{\theta}) + \frac{1-t}{t} \nabla_{\boldsymbol{x}_t} \log p_{\boldsymbol{\beta}}(y \mid \boldsymbol{x}_t, t) \end{aligned} \tag{16}$$

## A.2 EXTENDED COMPARISONS

We have expanded our analysis to include widely recognized methods such as NSGD-III Deb & Jain (2013) and SMS-EMOA Beume et al. (2007), applied to the same five tasks highlighted in our ablation studies. Our findings in Table 3 demonstrate that ParetoFlow consistently outperforms these traditional approaches, reinforcing the effectiveness and robustness of our method.

Table 3: Comparison of NSGD-III and SMS-EMOA

| Methods | ZDT2 | C-10/MOP1 | MO-Hopper | Zinc | RE23 |
|---|---|---|---|---|---|
| NSGD-III + MM | $5.74 \pm 0.05$ | $4.71 \pm 0.01$ | $5.31 \pm 0.13$ | $4.15 \pm 0.06$ | $4.96 \pm 0.04$ |
| SMS-EMOA + MM | $6.23 \pm 0.09$ | $4.73 \pm 0.00$ | $5.45 \pm 0.23$ | $4.33 \pm 0.09$ | $5.67 \pm 0.08$ |
| ParetoFlow (ours) | $\mathbf{6.79 \pm 0.16}$ | $\mathbf{4.77 \pm 0.00}$ | $\mathbf{5.69 \pm 0.03}$ | $\mathbf{4.49 \pm 0.06}$ | $\mathbf{6.32 \pm 0.46}$ |

## A.3 HYPERPARAMETER SENSITIVITY

This section examines the sensitivity of our method to various hyperparameters—namely, the number of neighbors ($K$), the number of offspring ($O$), the number of sampling steps ($t$), the scaling factor ($\gamma$) in Eq.(8), the noise factor ($g$) in Eq.(10)—across two tasks: the continuous MO-Hopper

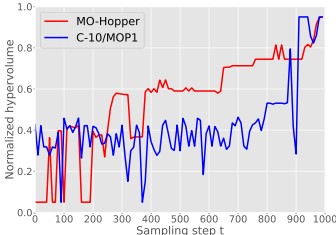
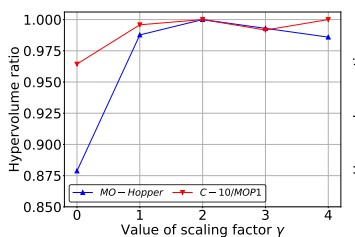
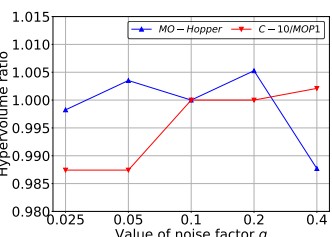

Figure 6: Hypervolume as a function of $t$.

Figure 7: Sensitivity to the scaling factor of $\gamma$.

Figure 8: Sensitivity to the noise factor $g$.

and the discrete C-10/MOP1. Hypervolume metrics are normalized by dividing by the default hyperparameter result to facilitate comparative analysis, unless specified otherwise.

**Number of Neighbors** ($K$)**:** Tested values include 1, 2, 3, 4, and 5, with $K = 3$ as the default. As shown in Figure 4, performance remains stable with changes in $K$. While performance generally increases with $K$, indicating more neighbors provide more useful information, it stops the increase at $K = 4$, likely limited by predictor accuracy and redundancy at higher neighbor counts.

**Number of Offspring** ($O$)**:** We vary the number of offspring, testing values of 3, 4, 5, 6, and 7, with $O = 5$ as the default. As illustrated in Figure 5, performance is stable across different $O$ values. Performance tends to increase with larger $O$, as more offspring provide additional options for subsequent iterations; however, this benefit is offset by increased computational costs.

**Number of Sampling Steps** ($t$)**:** We analyze the impact of the number of sampling steps $t$ on our method's effectiveness. The normalized hypervolume of the Pareto set ($PS$) is plotted as a function of time step $t$ in Figure 6. We observe a general increase in hypervolume with increasing $t$. Additionally, the robustness of our method to changes in $T$ is further examined in the Appendix A.11.

**Scaling Factor** ($\gamma$)**:** The effect of varying $\gamma$ is investigated with values 0, 1, 2, 3, and 4, and $\gamma = 2$ as the standard setting. As indicated in Figure 7, performance remains stable across changes in $\gamma$, and improves from $\gamma = 0$ to $\gamma = 2$, demonstrating the effectiveness of increasing predictor guidance.

**Noise Factor** ($g$)**:** The effect of varying $g$ is investigated with values 0.025, 0.05, 0.1, 0.2, and 0.4, and $g = 0.1$ as the default. As shown in Figure 8, performance is quite robust to changes in $g$.

### A.4 TRAINING DETAILS

We adopt the predictor training configurations from Xue et al. (2024), utilizing a multiple model setup. Each predictor consists of a 3-layer MLP with ReLU activations, featuring a hidden layer size of 2048. These models are trained over 200 epochs with a batch size of 128, using the Adam optimizer Kingma & Ba (2015) at an initial learning rate of $1 \times 10^{-3}$, and a decay rate of 0.98 per epoch. Flow matching training follows protocols from Tomczak (2022), employing a 4-layer MLP with SeLU activations and a hidden layer size of 512. The model undergoes 1000 training epochs with early stopping after 20 epochs of no improvement, with a batch size of 128 and the Adam optimizer.

The approximation $\hat{\boldsymbol{x}}_1(\boldsymbol{x}_t)$ proves inaccurate when $t$ is near 0, leading to an unreliable predictor. Consequently, we set $\gamma = 2$ only if $t$ exceeds a predefined threshold; otherwise, $\gamma = 0$. We determine this threshold by evaluating the reconstruction loss between $\hat{\boldsymbol{x}}_1(\boldsymbol{x}_t)$ and $\boldsymbol{x}_1$. As illustrated in Figure 9 for the tasks C-10/MOP1 and MO-Hopper, the reconstruction loss remains below 0.2 when $t$ exceeds 0.8. Therefore, we establish the threshold at 0.8.

We employ the simplest setup of Multiple Models (MM) within *ParetoFlow*. As detailed in Table 4, we experiment with the IOM setup for comparison. The results indicate that the IOM setup performs similarly or slightly worse than the MM setup.

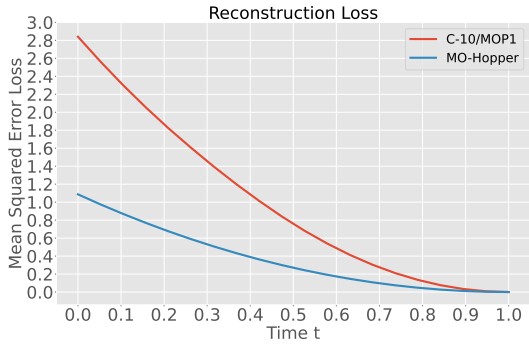
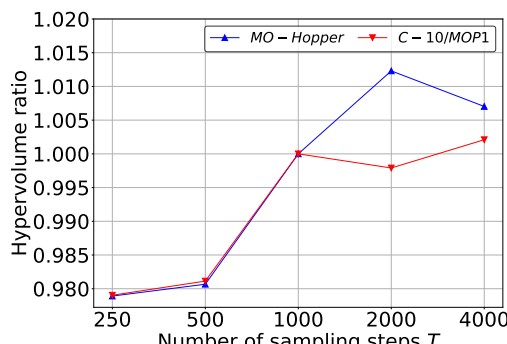

Figure 9: Reconstruction loss as a function of the time step $t$.

Figure 10: Sensitivity to the number of sampling steps $T$.

Table 4: Comparison of ParetoFlow with Vanilla Multiple Models and with IOM Multiple Models.

| Methods | ZDT2 | C-10/MOP1 | MO-Hopper | Zinc | RE23 |
|---|---|---|---|---|---|
| ParetoFlow (default) | $\mathbf{6.79 \pm 0.16}$ | $\mathbf{4.77 \pm 0.00}$ | $5.69 \pm 0.03$ | $\mathbf{4.49 \pm 0.06}$ | $\mathbf{6.32 \pm 0.46}$ |
| ParetoFlow + IOM | $6.32 \pm 0.22$ | $4.75 \pm 0.01$ | $\mathbf{5.97 \pm 0.13}$ | $4.45 \pm 0.04$ | $6.15 \pm 0.14$ |

## A.5 COMPUTATIONAL COST

All experiments are conducted on a workstation with an Intel i9-12900K CPU and an NVIDIA RTX3090 GPU. The computational cost of our method consists of three components: predictor training, flow model training, and design sampling. Detailed task information and time costs are summarized in Table 5 (in minutes). For discrete tasks, we report the dimension of the converted logits. Our method is efficient, completing most tasks within 10 minutes.

We also benchmark some baseline methods in Table 6. Considering the minimal overhead and significant performance gains of our method compared to baselines, we believe it provides a practical solution for researchers and practitioners seeking effective results without sacrificing speed.

Table 5: Time cost of ParetoFlow.

| Components | ZDT2 | C-10/MOP1 | MO-Hopper | Zinc | RE23 |
|---|---|---|---|---|---|
| Design dimension | 30 | 31 | 10184 | 378 | 4 |
| Number of offline points | 60000 | 12084 | 4500 | 48000 | 60000 |
| Number of objectives | 2 | 2 | 2 | 2 | 2 |
| Predictor training (min) | 3.99 | 0.76 | 1.03 | 3.21 | 3.94 |
| Flow model training (min) | 2.36 | 1.28 | 2.51 | 6.56 | 1.58 |
| Design sampling (min) | 0.55 | 0.25 | 0.64 | 0.42 | 0.63 |
| Overall time cost (min) | 6.90 | 2.29 | 4.18 | 10.19 | 6.15 |

Table 6: Time cost of baselines.

| Tasks | C-10/MOP1 | MO-Hopper |
|---|---|---|
| E2E | 1.20 | 1.17 |
| MH | 1.24 | 1.17 |
| MM | 1.70 | 1.80 |
| MOEA/D + MM | 1.50 | 1.36 |
| MOBO | 0.12 | 33.68 |
| PROUD | 2.23 | 4.05 |
| LaMBO-2 | 2.54 | 4.22 |
| CorrVAE | 1.81 | 2.89 |
| MOGFN | 5.73 | 10.62 |
| ParetoFlow (ours) | 2.29 | 4.18 |

To provide more detailed insights, we have conducted a thorough analysis across a diverse set of MO-NAS tasks from the NAS-Bench series, including C-10/MOP1, MOP2, MOP5, MOP6, and MOP7, as detailed in Table 7. Our findings indicate that despite variations in the number of objectives and design dimensions, the computational cost of our method remains consistent. Notably, for tasks with higher dimensional objectives such as MO-Swimmer and MO-Hopper, our computational efficiency is comparable to tasks with lower dimensions. This consistency underscores our method's scalability across a range of computational demands. Additionally, our method consistently achieves strong performance, further attesting to its robustness.

Table 7: Task-Specific Details

| Tasks | C-10/MOP1 | C-10/MOP2 | C-10/MOP5 | C-10/MOP6 | C-10/MOP7 | MO-Hopper | MO-Swimmer |
|---|---|---|---|---|---|---|---|
| Raw Input Dimension | 26 | 26 | 6 | 6 | 6 | 10184 | 9734 |
| Input Dimension (Logits) | 31 | 31 | 24 | 24 | 24 | $N/A$ | $N/A$ |
| Number of Objectives | 2 | 3 | 5 | 6 | 8 | 2 | 2 |
| Number of Offline Samples | 12084 | 26316 | 9375 | 9375 | 9375 | 4500 | 8571 |
| Predictor Training (min) | 0.76 | 2.44 | 1.45 | 1.80 | 2.38 | 1.03 | 1.84 |
| Flow Model Training (min) | 1.28 | 5.92 | 1.20 | 5.18 | 1.19 | 2.51 | 2.53 |
| Design Sampling (min) | 0.25 | 0.52 | 0.41 | 0.44 | 0.51 | 0.64 | 0.60 |
| Overall Time Cost (min) | 2.29 | 8.88 | 3.06 | 7.42 | 4.08 | 4.18 | 4.97 |
| Rank of ParetoFlow | 1 | 1 | 1 | 1 | 1 | 1 | 1 |

## A.6 100TH PERCENTILE RESULTS

As shown in Tables 8, 9, 10, 11, 12, and 13, we present the 100th percentile results with 256 solutions, demonstrating that our method, *ParetoFlow*, performs well across different tasks. For each task, algorithms within one standard deviation of having the highest performance are **bolded**.

Table 8: Hypervolume results for synthetic functions.

| Methods | DTLZ1 | DTLZ7 | OmniTest | VLMOP1 | VLMOP2 | VLMOP3 | ZDT1 | ZDT2 | ZDT3 | ZDT4 | ZDT6 |
|---|---|---|---|---|---|---|---|---|---|---|---|
| $\mathcal{D}$(best) | 10.43 | 8.32 | 3.87 | 0.08 | 1.64 | 45.14 | 4.04 | 4.70 | 5.05 | 5.46 | 4.76 |
| E2E | $10.12 \pm 0.02$ | $10.70 \pm 0.01$ | $4.35 \pm 0.00$ | $2.57 \pm 2.26$ | $4.24 \pm 0.01$ | $46.93 \pm 0.00$ | $2.69 \pm 0.00$ | $3.21 \pm 0.00$ | $5.50 \pm 0.04$ | $3.12 \pm 0.09$ | $\mathbf{4.92 \pm 0.00}$ |
| E2E + GradNorm | $10.65 \pm 0.00$ | $10.71 \pm 0.00$ | $3.76 \pm 0.03$ | $2.33 \pm 2.33$ | $2.79 \pm 1.34$ | $42.23 \pm 0.98$ | $4.77 \pm 0.01$ | $5.63 \pm 0.02$ | $5.27 \pm 0.03$ | $3.23 \pm 0.03$ | $3.81 \pm 1.02$ |
| E2E + PcGrad | $\mathbf{10.65 \pm 0.00}$ | $10.52 \pm 0.00$ | $4.35 \pm 0.00$ | $2.57 \pm 2.26$ | $4.14 \pm 0.07$ | $46.79 \pm 0.06$ | $4.84 \pm 0.01$ | $5.70 \pm 0.01$ | $5.45 \pm 0.00$ | $3.12 \pm 0.01$ | $2.04 \pm 0.22$ |
| MH | $10.38 \pm 0.25$ | $10.63 \pm 0.11$ | $4.30 \pm 0.05$ | $2.57 \pm 2.26$ | $4.26 \pm 0.00$ | $46.92 \pm 0.02$ | $2.69 \pm 0.00$ | $4.48 \pm 1.27$ | $5.50 \pm 0.04$ | $3.23 \pm 0.16$ | $4.91 \pm 0.00$ |
| MH + GradNorm | $\mathbf{10.65 \pm 0.22}$ | $10.61 \pm 0.10$ | $4.34 \pm 0.00$ | $0.00 \pm 0.00$ | $4.13 \pm 0.03$ | $46.64 \pm 0.22$ | $4.83 \pm 0.00$ | $5.68 \pm 0.05$ | $5.26 \pm 0.04$ | $3.39 \pm 0.00$ | $4.87 \pm 0.00$ |
| MH + PcGrad | $10.64 \pm 0.00$ | $10.49 \pm 0.01$ | $4.35 \pm 0.00$ | $2.55 \pm 2.24$ | $4.01 \pm 0.02$ | $46.91 \pm 0.00$ | $2.73 \pm 0.03$ | $5.69 \pm 0.03$ | $5.45 \pm 0.00$ | $3.64 \pm 0.17$ | $2.17 \pm 0.05$ |
| MM | $\mathbf{10.65 \pm 0.00}$ | $\mathbf{10.73 \pm 0.00}$ | $4.35 \pm 0.00$ | $2.57 \pm 2.26$ | $\mathbf{4.28 \pm 0.00}$ | $\mathbf{46.94 \pm 0.00}$ | $4.75 \pm 0.00$ | $5.58 \pm 0.00$ | $\mathbf{5.80 \pm 0.01}$ | $4.14 \pm 0.20$ | $4.91 \pm 0.00$ |
| MM + COMs | $10.64 \pm 0.01$ | $9.64 \pm 0.22$ | $4.29 \pm 0.03$ | $2.54 \pm 2.25$ | $1.90 \pm 0.05$ | $46.78 \pm 0.07$ | $4.24 \pm 0.01$ | $4.89 \pm 0.07$ | $5.54 \pm 0.02$ | $4.56 \pm 0.04$ | $4.57 \pm 0.00$ |
| MM + RoMA | $10.64 \pm 0.01$ | $10.63 \pm 0.03$ | $3.03 \pm 0.03$ | $2.54 \pm 2.24$ | $1.46 \pm 0.00$ | $44.15 \pm 2.36$ | $\mathbf{4.87 \pm 0.00}$ | $5.65 \pm 0.00$ | $5.78 \pm 0.02$ | $3.18 \pm 0.05$ | $1.77 \pm 0.02$ |
| MM + IOM | $\mathbf{10.65 \pm 0.00}$ | $\mathbf{10.74 \pm 0.08}$ | $4.34 \pm 0.00$ | $2.55 \pm 2.24$ | $3.77 \pm 0.01$ | $46.92 \pm 0.00$ | $4.66 \pm 0.01$ | $5.74 \pm 0.01$ | $5.61 \pm 0.01$ | $4.65 \pm 0.19$ | $4.89 \pm 0.02$ |
| MM + ICT | $10.64 \pm 0.00$ | $\mathbf{10.75 \pm 0.02}$ | $4.30 \pm 0.00$ | $0.26 \pm 0.06$ | $1.46 \pm 0.00$ | $46.74 \pm 0.09$ | $4.39 \pm 0.01$ | $5.53 \pm 0.00$ | $4.37 \pm 0.03$ | $3.44 \pm 0.16$ | $2.33 \pm 0.11$ |
| MM + Tri-Mentor | $10.64 \pm 0.00$ | $10.67 \pm 0.01$ | $3.97 \pm 0.00$ | $\mathbf{4.83 \pm 0.00}$ | $1.46 \pm 0.00$ | $46.82 \pm 0.02$ | $4.52 \pm 0.02$ | $5.55 \pm 0.01$ | $5.62 \pm 0.09$ | $3.47 \pm 0.04$ | $2.36 \pm 0.28$ |
| MOEA/D + MM | $10.64 \pm 0.00$ | $10.36 \pm 0.02$ | $4.77 \pm 0.00$ | $0.31 \pm 0.02$ | $4.01 \pm 0.01$ | $45.49 \pm 0.10$ | $4.44 \pm 0.04$ | $5.29 \pm 0.05$ | $5.38 \pm 0.08$ | $4.87 \pm 0.15$ | $4.78 \pm 0.01$ |
| MOBO | $\mathbf{10.65 \pm 0.00}$ | $10.51 \pm 0.05$ | $4.35 \pm 0.00$ | $0.32 \pm 0.00$ | $2.18 \pm 0.69$ | $46.91 \pm 0.03$ | $4.44 \pm 0.09$ | $5.18 \pm 0.09$ | $5.41 \pm 0.12$ | $4.60 \pm 0.13$ | $3.96 \pm 0.73$ |
| MOBO-$q$ParEGO | $10.63 \pm 0.01$ | $10.25 \pm 0.05$ | $4.33 \pm 0.00$ | $0.29 \pm 0.01$ | $2.93 \pm 0.06$ | $46.93 \pm 0.00$ | $4.32 \pm 0.02$ | $5.12 \pm 0.17$ | $5.20 \pm 0.01$ | $4.81 \pm 0.10$ | $3.31 \pm 0.03$ |
| MOBO-JES | $10.61 \pm 0.00$ | $9.36 \pm 0.08$ | $3.87 \pm 0.00$ | N/A | $1.46 \pm 0.00$ | $46.88 \pm 0.00$ | $3.97 \pm 0.09$ | $4.44 \pm 0.07$ | $5.17 \pm 0.02$ | $4.43 \pm 0.08$ | $3.09 \pm 0.02$ |
| PROUD | $10.61 \pm 0.01$ | $9.16 \pm 0.01$ | $\mathbf{4.78 \pm 0.00}$ | $3.12 \pm 0.35$ | $4.01 \pm 0.01$ | $\mathbf{46.94 \pm 0.00}$ | $4.20 \pm 0.04$ | $6.32 \pm 0.07$ | $5.23 \pm 0.07$ | $4.92 \pm 0.05$ | $4.50 \pm 0.03$ |
| LaMBO-2 | $10.62 \pm 0.01$ | $9.21 \pm 0.06$ | $\mathbf{4.78 \pm 0.00}$ | $3.08 \pm 0.30$ | $4.01 \pm 0.02$ | $46.67 \pm 0.03$ | $4.18 \pm 0.05$ | $6.36 \pm 0.23$ | $5.14 \pm 0.13$ | $4.92 \pm 0.14$ | $4.51 \pm 0.15$ |
| CorrVAE | $10.60 \pm 0.01$ | $9.13 \pm 0.03$ | $4.68 \pm 0.00$ | $3.04 \pm 0.16$ | $4.00 \pm 0.01$ | $46.93 \pm 0.00$ | $4.16 \pm 0.03$ | $6.21 \pm 0.07$ | $5.14 \pm 0.07$ | $4.85 \pm 0.07$ | $4.43 \pm 0.09$ |
| MOGFN | $10.61 \pm 0.01$ | $9.15 \pm 0.02$ | $4.77 \pm 0.00$ | $3.48 \pm 0.06$ | $4.01 \pm 0.01$ | $46.79 \pm 0.03$ | $4.17 \pm 0.03$ | $6.27 \pm 0.08$ | $5.19 \pm 0.05$ | $4.90 \pm 0.05$ | $4.48 \pm 0.04$ |
| ParetoFlow (**ours**) | $\mathbf{10.65 \pm 0.00}$ | $10.60 \pm 0.03$ | $\mathbf{4.78 \pm 0.00}$ | $3.15 \pm 0.28$ | $4.20 \pm 0.02$ | $\mathbf{46.94 \pm 0.00}$ | $\mathbf{4.30 \pm 0.02}$ | $\mathbf{6.79 \pm 0.16}$ | $\mathbf{5.82 \pm 0.03}$ | $\mathbf{5.15 \pm 0.08}$ | $4.62 \pm 0.04$ |

Table 9: Hypervolume results for MO-NAS (Part 1).

| Methods | C-10/MOP1 | C-10/MOP2 | C-10/MOP3 | C-10/MOP4 | C-10/MOP5 | C-10/MOP6 | C-10/MOP7 | C-10/MOP8 | C-10/MOP9 |
|---|---|---|---|---|---|---|---|---|---|
| $\mathcal{D}$(best) | 4.72 | 10.42 | 9.21 | 18.62 | 40.79 | 103.55 | 399.67 | 4.38 | 9.64 |
| E2E | 4.74 ± 0.01 | 10.45 ± 0.01 | 10.15 ± 0.00 | **21.47 ± 0.42** | 49.17 ± 0.02 | 105.52 ± 0.33 | 491.95 ± 0.58 | 4.62 ± 0.08 | **10.34 ± 0.04** |
| E2E + GradNorm | 4.63 ± 0.01 | 10.47 ± 0.01 | 9.26 ± 0.00 | 19.08 ± 0.38 | 48.94 ± 0.01 | 102.49 ± 1.59 | 487.55 ± 1.04 | 3.94 ± 0.04 | 9.95 ± 0.04 |
| E2E + PcGrad | 4.76 ± 0.01 | **10.48 ± 0.02** | 10.16 ± 0.02 | 21.35 ± 0.37 | 49.24 ± 0.01 | **106.14 ± 0.40** | **494.60 ± 0.00** | 4.61 ± 0.04 | 10.09 ± 0.17 |
| MH | 4.74 ± 0.00 | **10.49 ± 0.03** | 10.09 ± 0.04 | **21.62 ± 0.09** | 49.14 ± 0.05 | 104.55 ± 1.38 | **496.05 ± 0.82** | 4.63 ± 0.05 | 9.88 ± 0.22 |
| MH + GradNorm | 4.74 ± 0.00 | 10.20 ± 0.01 | 9.28 ± 0.06 | 18.53 ± 1.21 | 47.26 ± 1.883 | 76.66 ± 12.15 | 389.61 ± 34.73 | 3.89 ± 0.20 | 9.54 ± 0.76 |
| MH + PcGrad | 4.74 ± 0.02 | 10.45 ± 0.03 | 10.01 ± 0.03 | 21.14 ± 0.28 | 49.20 ± 0.01 | **106.57 ± 0.21** | 492.25 ± 2.69 | 4.59 ± 0.06 | 10.17 ± 0.00 |
| MM | 4.73 ± 0.00 | 10.44 ± 0.00 | **10.18 ± 0.00** | 21.23 ± 0.06 | 48.82 ± 0.00 | 104.91 ± 0.01 | 493.33 ± 0.00 | 4.58 ± 0.00 | 10.15 ± 0.00 |
| MM + COMs | 4.76 ± 0.00 | 10.44 ± 0.00 | 10.13 ± 0.00 | 20.91 ± 0.00 | 48.90 ± 0.00 | 106.00 ± 0.16 | 491.70 ± 0.00 | 4.55 ± 0.00 | 10.12 ± 0.00 |
| MM + RoMA | **4.77 ± 0.00** | 10.46 ± 0.00 | 10.16 ± 0.00 | **21.67 ± 0.25** | 48.96 ± 0.00 | 105.75 ± 0.57 | 485.46 ± 0.00 | 4.35 ± 0.00 | 9.76 ± 0.02 |
| MM + IOM | 4.75 ± 0.01 | 10.46 ± 0.01 | 10.07 ± 0.00 | **21.59 ± 0.33** | 49.20 ± 0.00 | **106.37 ± 0.05** | 490.47 ± 0.00 | 4.66 ± 0.00 | **10.33 ± 0.00** |
| MM + ICT | 4.74 ± 0.01 | 10.46 ± 0.00 | 9.96 ± 0.00 | 20.60 ± 0.00 | 49.17 ± 0.09 | **106.29 ± 0.01** | 491.90 ± 0.00 | 4.62 ± 0.00 | 9.62 ± 0.00 |
| MM + Tri-Mentor | **4.77 ± 0.00** | 10.46 ± 0.00 | 10.15 ± 0.00 | 21.58 ± 0.08 | 45.46 ± 0.00 | **106.41 ± 0.01** | 491.88 ± 0.00 | 4.61 ± 0.02 | 8.84 ± 0.21 |
| MOEA/D + MM | 4.74 ± 0.03 | 9.87 ± 0.05 | 9.80 ± 0.17 | 21.30 ± 0.21 | 48.84 ± 0.10 | 105.89 ± 0.18 | 492.23 ± 2.69 | 4.19 ± 0.03 | 9.62 ± 0.02 |
| MOBO | 4.74 ± 0.00 | 10.43 ± 0.01 | 8.58 ± 0.00 | 20.35 ± 0.02 | 44.89 ± 0.01 | 102.33 ± 0.07 | 488.97 ± 4.01 | 4.32 ± 0.00 | 8.77 ± 0.03 |
| MOBO-qParEGO | 4.63 ± 0.01 | 10.44 ± 0.00 | 8.94 ± 0.05 | 20.01 ± 0.05 | 37.21 ± 0.00 | 94.72 ± 5.91 | 350.55 ± 0.13 | 4.50 ± 0.00 | 8.36 ± 0.02 |
| MOBO-JES | N/A | N/A | N/A | N/A | N/A | N/A | N/A | N/A | N/A |
| PROUD | 4.71 ± 0.04 | 10.46 ± 0.00 | 8.66 ± 0.11 | 18.41 ± 0.59 | 47.26 ± 0.21 | 101.86 ± 1.11 | 466.36 ± 11.76 | 4.14 ± 0.05 | 9.54 ± 0.13 |
| LaMBO-2 | 4.73 ± 0.03 | 10.46 ± 0.04 | 8.54 ± 0.27 | 18.33 ± 0.58 | 47.22 ± 0.40 | 102.33 ± 1.82 | 476.51 ± 12.54 | 4.10 ± 0.13 | 9.42 ± 0.18 |
| CorrVAE | 4.68 ± 0.03 | 10.43 ± 0.02 | 8.40 ± 0.12 | 17.53 ± 0.72 | 47.00 ± 0.19 | 100.62 ± 0.44 | 454.27 ± 6.26 | 4.04 ± 0.10 | 9.36 ± 0.13 |
| MOGFN | 4.70 ± 0.02 | 10.45 ± 0.00 | 8.46 ± 0.11 | 17.94 ± 0.34 | 47.12 ± 0.11 | 101.24 ± 0.97 | 459.87 ± 10.01 | 4.11 ± 0.05 | 9.45 ± 0.11 |
| ParetoFlow (ours) | **4.77 ± 0.00** | **10.50 ± 0.02** | 9.76 ± 0.14 | 20.98 ± 0.12 | **50.14 ± 0.61** | **106.58 ± 0.51** | **497.19 ± 3.14** | **4.68 ± 0.00** | 9.95 ± 0.02 |

Table 10: Hypervolume results for MO-NAS (Part 2).

| Methods | IN-1K/MOP1 | IN-1K/MOP2 | IN-1K/MOP3 | IN-1K/MOP4 | IN-1K/MOP5 | IN-1K/MOP6 | IN-1K/MOP7 | IN-1K/MOP8 | IN-1K/MOP9 | NasBench201-Test |
|---|---|---|---|---|---|---|---|---|---|---|
| $\mathcal{D}$(best) | 4.36 | 4.45 | 9.86 | 4.15 | 4.30 | 9.15 | 3.70 | 9.13 | 18.87 | 9.89 |
| E2E | 4.59 ± 0.04 | **4.56 ± 0.02** | 9.95 ± 0.02 | 4.51 ± 0.05 | **4.72 ± 0.09** | 9.91 ± 0.19 | 4.27 ± 0.19 | 9.52 ± 0.03 | 19.51 ± 0.34 | 9.14 ± 0.06 |
| E2E + GradNorm | 4.11 ± 0.06 | 4.47 ± 0.00 | 8.04 ± 0.01 | 4.46 ± 0.03 | 4.49 ± 0.01 | 9.59 ± 0.19 | 4.12 ± 0.09 | 8.44 ± 0.18 | 17.56 ± 0.25 | 8.98 ± 0.02 |
| E2E + PcGrad | 4.52 ± 0.03 | 4.49 ± 0.03 | **10.04 ± 0.00** | 4.40 ± 0.05 | 4.52 ± 0.05 | 9.81 ± 0.07 | 3.94 ± 0.19 | 9.52 ± 0.08 | 19.63 ± 0.21 | 9.17 ± 0.00 |
| MH | **4.61 ± 0.00** | 4.54 ± 0.05 | 9.98 ± 0.09 | 4.54 ± 0.05 | **4.70 ± 0.01** | **10.10 ± 0.07** | 4.27 ± 0.11 | 9.49 ± 0.05 | 20.20 ± 0.12 | 9.09 ± 0.09 |
| MH + GradNorm | 4.25 ± 0.01 | 3.94 ± 0.47 | 8.87 ± 1.03 | 4.43 ± 0.08 | 4.49 ± 0.05 | 9.58 ± 0.25 | 2.93 ± 0.27 | 5.21 ± 1.19 | 10.20 ± 2.16 | 9.03 ± 0.08 |
| MH + PcGrad | 4.57 ± 0.00 | 4.53 ± 0.01 | 10.00 ± 0.03 | 4.38 ± 0.05 | 4.48 ± 0.02 | 9.84 ± 0.12 | 4.04 ± 0.09 | 9.57 ± 0.06 | 19.89 ± 0.09 | 9.19 ± 0.01 |
| MM | 4.56 ± 0.00 | 4.54 ± 0.00 | **10.05 ± 0.00** | **4.59 ± 0.00** | 4.52 ± 0.00 | 9.85 ± 0.24 | 4.14 ± 0.03 | 9.56 ± 0.05 | 19.92 ± 0.42 | 9.19 ± 0.00 |
| MM + COMs | 4.26 ± 0.00 | 4.32 ± 0.00 | 8.02 ± 0.01 | 4.40 ± 0.01 | 4.46 ± 0.00 | 9.95 ± 0.05 | 3.98 ± 0.09 | 9.55 ± 0.02 | 20.07 ± 0.03 | 9.93 ± 0.01 |
| MM + RoMA | 4.60 ± 0.00 | 4.11 ± 0.05 | 8.33 ± 0.01 | 4.56 ± 0.10 | 4.42 ± 0.07 | 10.02 ± 0.07 | **4.47 ± 0.06** | 9.48 ± 0.01 | 19.87 ± 0.29 | 9.13 ± 0.00 |
| MM + IOM | **4.61 ± 0.00** | **4.57 ± 0.00** | 10.02 ± 0.00 | 4.43 ± 0.07 | 4.55 ± 0.00 | 9.65 ± 0.11 | 4.00 ± 0.04 | 9.67 ± 0.05 | 20.33 ± 0.02 | 9.15 ± 0.02 |
| MM + ICT | 4.56 ± 0.00 | 4.36 ± 0.00 | 9.60 ± 0.00 | 4.48 ± 0.02 | 4.49 ± 0.05 | 9.99 ± 0.01 | 3.96 ± 0.12 | 9.30 ± 0.29 | 19.08 ± 0.75 | 9.16 ± 0.02 |
| MM + Tri-Mentor | 4.32 ± 0.01 | 4.45 ± 0.07 | 9.81 ± 0.00 | 4.32 ± 0.05 | 4.42 ± 0.05 | 9.97 ± 0.28 | 4.23 ± 0.03 | 9.57 ± 0.00 | 15.28 ± 1.61 | 9.17 ± 0.00 |
| MOEA/D + MM | 4.13 ± 0.01 | 4.47 ± 0.05 | 9.69 ± 0.03 | 4.27 ± 0.04 | 4.37 ± 0.09 | 9.98 ± 0.07 | 3.98 ± 0.05 | 8.58 ± 0.06 | 16.49 ± 2.32 | 8.35 ± 0.03 |
| MOBO | 4.26 ± 0.04 | 4.53 ± 0.02 | 8.24 ± 0.03 | 4.22 ± 0.05 | 4.30 ± 0.10 | 9.51 ± 0.06 | 3.99 ± 0.02 | 9.25 ± 0.14 | 18.27 ± 0.03 | 9.04 ± 0.01 |
| MOBO-qParEGO | 3.93 ± 0.06 | 4.28 ± 0.01 | 8.33 ± 0.14 | 4.18 ± 0.00 | 4.44 ± 0.09 | 9.52 ± 0.04 | 4.05 ± 0.08 | 8.67 ± 0.12 | 16.23 ± 0.05 | 9.05 ± 0.04 |
| MOBO-JES | N/A | N/A | N/A | N/A | N/A | N/A | N/A | N/A | N/A | 8.12 ± 0.05 |
| PROUD | 4.37 ± 0.01 | 4.23 ± 0.04 | 9.35 ± 0.09 | 3.96 ± 0.11 | 4.10 ± 0.13 | 9.29 ± 0.13 | 3.73 ± 0.04 | 8.91 ± 0.04 | 18.68 ± 0.05 | 10.08 ± 0.06 |
| LaMBO-2 | 4.39 ± 0.02 | 4.19 ± 0.05 | 9.43 ± 0.11 | 4.08 ± 0.07 | 4.10 ± 0.13 | 9.53 ± 0.20 | 3.76 ± 0.05 | 8.88 ± 0.03 | 18.73 ± 0.03 | **10.15 ± 0.07** |
| CorrVAE | 4.36 ± 0.02 | 4.18 ± 0.04 | 9.27 ± 0.05 | 3.93 ± 0.09 | 4.10 ± 0.13 | 9.16 ± 0.08 | 3.65 ± 0.07 | 8.86 ± 0.02 | 18.63 ± 0.03 | 9.17 ± 0.13 |
| MOGFN | 4.37 ± 0.01 | 4.21 ± 0.05 | 9.31 ± 0.05 | 3.98 ± 0.09 | 4.18 ± 0.08 | 9.22 ± 0.09 | 3.69 ± 0.04 | 8.87 ± 0.03 | 18.66 ± 0.04 | 10.04 ± 0.08 |
| ParetoFlow (ours) | **4.62 ± 0.01** | **4.58 ± 0.03** | **10.08 ± 0.04** | **4.63 ± 0.05** | **4.70 ± 0.10** | **10.05 ± 0.00** | 3.78 ± 0.06 | **9.79 ± 0.10** | **20.89 ± 0.06** | 9.36 ± 0.00 |

Table 11: Hypervolume results for MORL.

| Methods | MO-Hopper | MO-Swimmer |
|---|---|---|
| $\mathcal{D}$(best) | 4.21 | 2.85 |
| E2E | 4.76 ± 0.25 | 2.77 ± 0.03 |
| E2E + GradNorm | 5.02 ± 0.04 | 2.90 ± 0.07 |
| E2E + PcGrad | 4.60 ± 0.27 | 2.49 ± 0.05 |
| MH | 4.57 ± 0.28 | 2.91 ± 0.04 |
| MH + GradNorm | 3.78 ± 0.05 | 2.69 ± 0.24 |
| MH + PcGrad | 4.27 ± 0.61 | 2.49 ± 0.25 |
| MM | 4.58 ± 0.19 | 2.60 ± 0.15 |
| MM + COMs | 4.84 ± 0.17 | 2.71 ± 0.04 |
| MM + RoMA | 5.23 ± 0.23 | 2.78 ± 0.20 |
| MM + IOM | 5.32 ± 0.49 | 2.94 ± 0.11 |
| MM + ICT | 4.67 ± 0.12 | 3.11 ± 0.08 |
| MM + Tri-Mentor | 4.93 ± 0.11 | 2.82 ± 0.10 |
| MOEA/D + MM | 4.75 ± 0.28 | 2.86 ± 0.19 |
| MOBO | 4.43 ± 0.08 | 2.61 ± 0.02 |
| MOBO-qParEGO | N/A | N/A |
| MOBO-JES | N/A | N/A |
| PROUD | 5.65 ± 0.00 | 3.43 ± 0.04 |
| LaMBO-2 | 5.66 ± 0.03 | 3.41 ± 0.12 |
| CorrVAE | 5.64 ± 0.00 | 3.38 ± 0.08 |
| MOGFN | 5.54 ± 0.00 | 3.43 ± 0.04 |
| ParetoFlow (ours) | **5.69 ± 0.03** | **3.50 ± 0.07** |

Table 12: Hypervolume results for scientific design.

| Methods | Molecule | Regex | RFP | ZINC |
|---|---|---|---|---|
| $\mathcal{D}$(best) | 2.26 | 2.82 | 3.36 | 4.01 |
| E2E | 2.30 ± 0.48 | 2.80 ± 0.00 | 3.80 ± 0.04 | 4.17 ± 0.00 |
| E2E + GradNorm | 1.10 ± 0.03 | 2.80 ± 0.00 | 4.11 ± 0.30 | 4.17 ± 0.00 |
| E2E + PcGrad | 1.54 ± 0.53 | 2.80 ± 0.00 | 3.84 ± 0.05 | 4.16 ± 0.08 |
| MH | 2.08 ± 0.00 | 2.80 ± 0.00 | 3.75 ± 0.00 | 4.16 ± 0.00 |
| MH + GradNorm | 1.62 ± 0.61 | 2.38 ± 0.00 | 4.08 ± 0.32 | 4.21 ± 0.05 |
| MH + PcGrad | 1.22 ± 0.10 | 2.80 ± 0.00 | 4.19 ± 0.22 | 4.12 ± 0.02 |
| MM | 2.78 ± 0.00 | 2.80 ± 0.00 | 4.40 ± 0.02 | 4.16 ± 0.00 |
| MM + COMs | 2.30 ± 0.00 | 2.21 ± 0.17 | 4.14 ± 0.35 | 4.12 ± 0.05 |
| MM + RoMA | 1.65 ± 0.02 | 2.80 ± 0.00 | 4.13 ± 0.29 | 4.16 ± 0.01 |
| MM + IOM | 1.75 ± 0.33 | 2.80 ± 0.00 | 4.13 ± 0.28 | 4.17 ± 0.00 |
| MM + ICT | 1.37 ± 0.17 | 2.80 ± 0.00 | 4.41 ± 0.00 | 4.10 ± 0.07 |
| MM + Tri-Mentor | 2.03 ± 0.00 | 2.80 ± 0.00 | 4.12 ± 0.29 | 4.06 ± 0.01 |
| MOEA/D + MM | 1.47 ± 0.09 | 2.99 ± 0.00 | 3.96 ± 0.15 | 4.52 ± 0.05 |
| MOBO | 2.22 ± 0.08 | **5.12 ± 0.17** | 3.74 ± 0.00 | 4.26 ± 0.00 |
| MOBO-$q$ParEGO | 2.12 ± 0.04 | 4.26 ± 0.25 | 3.33 ± 0.00 | 4.05 ± 0.02 |
| MOBO-JES | 2.10 ± 1.04 | N/A | N/A | N/A |
| PROUD | 1.96 ± 0.48 | 3.26 ± 0.00 | 4.22 ± 0.25 | 4.37 ± 0.03 |
| LaMBO-2 | 2.18 ± 0.63 | 3.26 ± 0.00 | 4.25 ± 0.28 | 4.35 ± 0.06 |
| CorrVAE | 1.71 ± 0.06 | 3.26 ± 0.00 | 4.19 ± 0.10 | 4.33 ± 0.03 |
| MOGFN | 1.76 ± 0.07 | 3.26 ± 0.00 | **4.44 ± 0.02** | 4.36 ± 0.02 |
| ParetoFlow (**ours**) | **2.91 ± 0.11** | 3.96 ± 0.00 | 4.23 ± 0.09 | **4.49 ± 0.06** |

Table 13: Hypervolume results for RE.

| Methods | RE21 | RE22 | RE23 | RE24 | RE25 | RE31 | RE32 | RE33 | RE34 | RE35 | RE36 | RE37 | RE41 | RE42 | RE61 | MO-Portfolio |
|---|---|---|---|---|---|---|---|---|---|---|---|---|---|---|---|---|
| $\mathcal{D}$(best) | 4.10 | 4.78 | 4.75 | 4.59 | 4.79 | 10.23 | 10.53 | 10.59 | 9.30 | 10.08 | 7.61 | 4.72 | 18.27 | 14.52 | 97.49 | 3.78 |
| E2E | **4.60 ± 0.00** | 4.84 ± 0.00 | 4.84 ± 0.00 | 4.38 ± 0.00 | 4.84 ± 0.00 | 10.56 ± 0.00 | 10.64 ± 0.00 | 10.69 ± 0.00 | 10.11 ± 0.01 | 10.35 ± 0.01 | 10.22 ± 0.07 | 6.21 ± 0.00 | **20.41 ± 0.25** | 22.32 ± 0.29 | 109.13 ± 0.09 | 3.07 ± 0.16 |
| E2E + GradNorm | 4.57 ± 0.01 | 4.84 ± 0.00 | 2.64 ± 0.00 | 4.38 ± 0.00 | 4.84 ± 0.00 | 10.65 ± 0.00 | 10.63 ± 0.00 | 9.90 ± 0.00 | 9.17 ± 0.85 | 10.35 ± 0.01 | 4.59 ± 3.24 | 6.22 ± 0.01 | 19.62 ± 0.08 | 19.12 ± 1.69 | 109.03 ± 0.05 | 3.28 ± 0.15 |
| E2E + PcGrad | **4.60 ± 0.00** | 4.84 ± 0.00 | 4.84 ± 0.00 | 4.38 ± 0.00 | 4.60 ± 0.24 | 10.65 ± 0.00 | 10.63 ± 0.00 | 10.41 ± 0.07 | 10.10 ± 0.01 | 10.52 ± 0.07 | 9.89 ± 0.17 | 5.52 ± 0.00 | **20.65 ± 0.19** | 22.09 ± 0.36 | 108.97 ± 0.06 | 3.08 ± 0.05 |
| MH | 4.60 ± 0.15 | 4.84 ± 0.00 | 4.74 ± 0.00 | 4.78 ± 0.00 | 4.60 ± 0.24 | 10.65 ± 0.00 | 10.64 ± 0.00 | 10.69 ± 0.00 | 10.10 ± 0.01 | 10.42 ± 0.12 | 10.19 ± 0.07 | 5.78 ± 0.05 | **20.57 ± 0.13** | 22.33 ± 0.46 | 109.17 ± 0.06 | 3.18 ± 0.04 |
| MH + GradNorm | 4.12 ± 0.43 | 4.83 ± 0.01 | 4.49 ± 0.09 | 2.64 ± 0.00 | 3.95 ± 0.00 | 10.65 ± 0.00 | 10.63 ± 0.00 | 5.85 ± 0.00 | 9.96 ± 0.09 | 10.18 ± 0.41 | 8.06 ± 1.77 | 6.36 ± 0.01 | 19.22 ± 2.00 | 14.78 ± 6.20 | 106.66 ± 0.21 | 3.11 ± 0.11 |
| MH + PcGrad | 4.60 ± 0.02 | 4.84 ± 0.00 | 4.27 ± 0.12 | 4.83 ± 0.00 | 4.35 ± 0.00 | 7.66 ± 0.00 | 10.08 ± 0.00 | 10.61 ± 0.00 | 10.11 ± 0.01 | 10.54 ± 0.02 | 9.60 ± 0.25 | 6.42 ± 0.00 | **20.73 ± 0.09** | 22.48 ± 0.25 | 109.15 ± 0.17 | 3.09 ± 0.13 |
| MM | **4.60 ± 0.00** | 4.84 ± 0.00 | 4.84 ± 0.00 | 4.82 ± 0.00 | 4.84 ± 0.00 | 10.65 ± 0.00 | 10.63 ± 0.00 | 10.67 ± 0.00 | 10.11 ± 0.01 | 10.57 ± 0.00 | **10.20 ± 0.01** | 6.49 ± 0.00 | **20.70 ± 0.03** | 22.65 ± 0.02 | 109.04 ± 0.03 | 3.69 ± 0.03 |
| MM + COMs | 4.32 ± 0.05 | 4.84 ± 0.00 | 4.79 ± 0.01 | 4.59 ± 0.00 | 4.84 ± 0.00 | 5.28 ± 5.28 | 10.64 ± 0.00 | 10.56 ± 0.03 | 9.92 ± 0.00 | 10.55 ± 0.01 | 9.32 ± 0.09 | 5.99 ± 0.03 | 20.22 ± 0.05 | 17.43 ± 0.80 | 107.31 ± 0.00 | 2.20 ± 0.02 |
| MM + RoMA | **4.60 ± 0.00** | 4.84 ± 0.00 | 4.84 ± 0.00 | 4.79 ± 0.02 | 4.69 ± 0.00 | 10.65 ± 0.00 | 10.65 ± 0.00 | 10.66 ± 0.00 | 9.92 ± 0.01 | 10.56 ± 0.01 | 9.78 ± 0.06 | 6.49 ± 0.01 | **20.43 ± 0.05** | 21.16 ± 0.13 | 108.26 ± 0.07 | 2.92 ± 0.02 |
| MM + IOM | **4.60 ± 0.00** | 4.84 ± 0.00 | 4.84 ± 0.00 | 4.84 ± 0.00 | 4.84 ± 0.00 | 10.65 ± 0.00 | 10.65 ± 0.00 | 10.68 ± 0.00 | 10.11 ± 0.01 | 10.56 ± 0.01 | 10.05 ± 0.27 | 6.54 ± 0.00 | **20.65 ± 0.00** | 22.34 ± 0.02 | 108.34 ± 0.04 | 2.93 ± 0.00 |
| MM + ICT | **4.60 ± 0.00** | 4.84 ± 0.00 | 2.77 ± 0.00 | 4.67 ± 0.00 | 4.84 ± 0.00 | 10.65 ± 0.00 | 2.77 ± 0.00 | 10.51 ± 0.00 | 10.09 ± 0.01 | 10.55 ± 0.01 | 10.11 ± 0.12 | 6.25 ± 0.07 | **20.63 ± 0.06** | 22.04 ± 0.08 | 108.34 ± 0.49 | 2.05 ± 0.10 |
| MM + Tri-Mentor | **4.60 ± 0.00** | 4.84 ± 0.00 | 2.76 ± 0.00 | 4.83 ± 0.00 | 4.70 ± 0.00 | 10.65 ± 0.00 | 10.65 ± 0.00 | 10.54 ± 0.00 | 10.09 ± 0.01 | 10.58 ± 0.00 | 9.97 ± 0.03 | 6.38 ± 0.06 | **20.63 ± 0.06** | 21.57 ± 0.20 | 108.38 ± 0.59 | 2.63 ± 0.12 |
| MOEA/D + MM | 4.31 ± 0.04 | 4.84 ± 0.00 | 4.84 ± 0.02 | 4.81 ± 0.05 | 4.35 ± 0.13 | 10.39 ± 0.04 | 10.49 ± 0.03 | 10.48 ± 0.02 | 9.62 ± 0.02 | 10.41 ± 0.08 | 10.15 ± 0.01 | 6.71 ± 0.12 | 21.24 ± 0.30 | 21.13 ± 0.24 | 109.24 ± 0.62 | 3.55 ± 0.17 |
| MOBO | 4.40 ± 0.08 | 4.84 ± 0.00 | 4.84 ± 0.00 | 4.83 ± 0.00 | 4.84 ± 0.00 | 10.19 ± 0.00 | 10.64 ± 0.01 | 10.69 ± 0.00 | 10.11 ± 0.00 | 10.68 ± 0.00 | 0.00 ± 0.00 | 6.60 ± 0.00 | 19.74 ± 0.03 | 15.82 ± 0.64 | N/A | 3.29 ± 0.02 |
| MOBO-$q$ParEGO | 4.35 ± 0.04 | 4.61 ± 0.00 | 4.84 ± 0.00 | 3.74 ± 0.00 | 4.71 ± 0.00 | 10.64 ± 0.01 | 9.77 ± 0.02 | 10.61 ± 0.03 | 9.83 ± 0.05 | 10.60 ± 0.00 | 0.00 ± 0.00 | 5.87 ± 0.05 | N/A | N/A | N/A | 3.15 ± 0.04 |
| MOBO-JES | 4.51 ± 0.03 | 4.84 ± 0.00 | 4.83 ± 0.00 | 4.82 ± 0.00 | 4.84 ± 0.00 | 10.28 ± 0.00 | 10.65 ± 0.00 | 10.61 ± 0.03 | 9.89 ± 0.00 | 10.52 ± 0.02 | 8.72 ± 0.10 | 6.20 ± 0.03 | N/A | N/A | N/A | 3.53 ± 0.07 |
| PROUD | 4.46 ± 0.06 | 4.85 ± 0.12 | 5.87 ± 0.13 | 5.86 ± 0.03 | 5.73 ± 0.52 | 10.63 ± 0.06 | 25.40 ± 5.25 | 10.99 ± 0.23 | 13.65 ± 0.16 | 11.98 ± 0.14 | 8.79 ± 0.08 | 7.29 ± 0.41 | 19.23 ± 0.39 | 44.24 ± 5.74 | 127.81 ± 5.62 | 4.19 ± 0.11 |
| LaMBO-2 | 4.41 ± 0.07 | 4.87 ± 0.16 | 5.95 ± 0.69 | 5.90 ± 0.11 | 5.78 ± 0.49 | 10.66 ± 0.07 | 20.11 ± 7.56 | 11.48 ± 0.45 | 13.49 ± 0.38 | 11.90 ± 0.11 | 8.72 ± 0.21 | 7.56 ± 0.67 | 19.43 ± 0.47 | 42.04 ± 4.07 | 116.18 ± 1.01 | 4.25 ± 0.04 |
| CorrVAE | 4.38 ± 0.04 | 4.82 ± 0.08 | 5.67 ± 0.16 | 5.81 ± 0.03 | 5.53 ± 0.26 | 10.61 ± 0.01 | 18.72 ± 5.91 | 10.69 ± 0.14 | 13.37 ± 0.26 | 11.87 ± 0.09 | 8.68 ± 0.08 | 6.77 ± 0.30 | 19.10 ± 0.29 | 39.99 ± 1.25 | 115.28 ± 10.63 | 4.16 ± 0.09 |
| MOGFN | 4.40 ± 0.03 | 4.86 ± 0.02 | 5.78 ± 0.10 | 5.83 ± 0.03 | 5.83 ± 0.44 | 10.64 ± 0.06 | 21.97 ± 6.06 | 10.84 ± 0.22 | 13.50 ± 0.24 | 11.93 ± 0.08 | 8.74 ± 0.07 | 7.06 ± 0.30 | 19.28 ± 0.22 | 37.55 ± 1.32 | 142.17 ± 8.24 | 4.22 ± 0.04 |
| ParetoFlow (**ours**) | 4.52 ± 0.04 | **4.99 ± 0.11** | **6.32 ± 0.46** | **5.97 ± 0.09** | **5.83 ± 0.44** | **10.74 ± 0.06** | **33.92 ± 3.47** | **11.75 ± 0.44** | **14.07 ± 0.35** | **12.08 ± 0.13** | **9.24 ± 0.15** | **8.00 ± 0.22** | **20.75 ± 0.44** | **56.99 ± 3.19** | **135.14 ± 2.39** | **4.28 ± 0.02** |

## A.7 50TH PERCENTILE RESULTS

We evaluate the 50th percentile performance of 256 solutions. As shown in Table 14, our method achieves the highest overall ranking based on the 50th percentile results. For each task, algorithms with performance within one standard deviation of the best are **bolded**. Detailed hypervolume results are presented in Tables 15, 16, 17, 18, 19, and 20.

Table 14: Average rank of 50th percentile results on each type of task in Off-MOO-Bench.

| Methods | Synthetic | MO-NAS | MORL | Sci-Design | RE | All Tasks |
|---|---|---|---|---|---|---|
| $\mathcal{D}$(best) | 13.18 ± 4.86 | 9.26 ± 4.93 | 5.00 ± 3.00 | 6.00 ± 2.92 | 14.06 ± 4.74 | 11.15 ± 5.48 |
| E2E | 12.64 ± 8.04 | 5.63 ± 3.33 | 14.50 ± 3.50 | 13.50 ± 3.50 | 8.75 ± 5.95 | 9.02 ± 6.30 |
| E2E + GradNorm | 14.36 ± 6.65 | 15.42 ± 4.97 | 9.50 ± 0.50 | 11.75 ± 1.30 | 12.88 ± 6.28 | 13.90 ± 5.74 |
| E2E + PcGrad | 11.64 ± 6.76 | 6.58 ± 4.11 | 16.50 ± 1.50 | 10.00 ± 4.85 | 10.25 ± 6.78 | 9.42 ± 6.17 |
| MH | 9.73 ± 7.62 | 5.84 ± 4.84 | 8.50 ± 5.50 | 10.00 ± 4.64 | 11.25 ± 6.08 | 8.75 ± 6.35 |
| MH + GradNorm | 12.45 ± 7.37 | 17.26 ± 4.73 | 16.00 ± 4.00 | 16.75 ± 3.70 | 16.75 ± 5.31 | 16.00 ± 5.78 |
| MH + PcGrad | 11.09 ± 4.48 | 8.05 ± 4.06 | 10.00 ± 1.00 | 16.00 ± 3.24 | 12.00 ± 5.39 | 10.60 ± 5.03 |
| MM | **4.91 ± 4.32** | 6.32 ± 4.10 | 15.00 ± 2.00 | 13.75 ± 3.11 | 6.75 ± 4.58 | **7.06 ± 4.95** |
| MM + COMs | 12.91 ± 4.03 | 10.32 ± 5.37 | 13.50 ± 1.50 | 11.00 ± 3.61 | 14.06 ± 4.31 | 12.19 ± 4.84 |
| MM + RoMA | 13.82 ± 6.58 | **5.53 ± 4.25** | 11.50 ± 4.50 | 11.25 ± 3.56 | 11.88 ± 5.18 | 9.90 ± 6.12 |
| MM + IOM | 6.18 ± 3.19 | 6.16 ± 4.22 | 12.00 ± 3.00 | 11.75 ± 3.56 | 8.81 ± 4.88 | 7.63 ± 4.58 |
| MM + ICT | 14.64 ± 4.58 | 10.58 ± 4.43 | 11.00 ± 6.00 | 13.75 ± 3.83 | 12.31 ± 6.65 | 12.23 ± 5.49 |
| MM + Tri-Mentor | 10.91 ± 5.60 | 11.68 ± 4.38 | 13.00 ± 2.00 | 13.00 ± 4.85 | 9.69 ± 6.06 | 11.02 ± 5.27 |
| MOEA/D + MM | 8.73 ± 5.40 | 10.79 ± 6.39 | 13.00 ± 7.00 | 10.25 ± 6.76 | 9.50 ± 5.99 | 10.00 ± 6.20 |
| MOBO | 12.73 ± 5.69 | 15.74 ± 3.75 | 18.50 ± 0.50 | 11.00 ± 6.04 | 14.88 ± 5.45 | 14.58 ± 5.18 |
| MOBO-$q$ParEGO | 12.91 ± 4.01 | 17.37 ± 3.42 | 21.00 ± 0.00 | 10.25 ± 6.30 | 18.94 ± 3.65 | 16.50 ± 4.84 |
| MOBO-JES | 17.00 ± 3.52 | 21.32 ± 2.90 | 21.00 ± 0.00 | 21.50 ± 0.87 | 16.12 ± 4.44 | 18.81 ± 4.22 |
| PROUD | 9.09 ± 4.76 | 14.05 ± 3.41 | 4.50 ± 2.50 | 3.75 ± 1.92 | 6.62 ± 5.58 | 9.56 ± 5.73 |
| LaMBO-2 | 9.73 ± 5.12 | 13.37 ± 4.29 | 4.00 ± 0.00 | 4.50 ± 1.80 | 7.69 ± 6.03 | 9.81 ± 5.76 |
| CorrVAE | 11.91 ± 5.25 | 17.42 ± 3.76 | 6.50 ± 1.50 | 5.75 ± 2.86 | 11.19 ± 5.50 | 13.02 ± 5.92 |
| MOGFN | 9.55 ± 5.35 | 15.47 ± 4.11 | 4.50 ± 1.50 | 4.00 ± 3.08 | 8.44 ± 5.07 | 10.75 ± 6.01 |
| ParetoFlow(**ours**) | 5.82 ± 3.76 | 6.63 ± 3.73 | **1.00 ± 0.00** | **2.50 ± 0.87** | **3.12 ± 3.76** | **4.85 ± 3.97** |

Table 15: Hypervolume results for synthetic functions.

| Methods | DTLZ1 | DTLZ7 | OmniTest | VLMOP1 | VLMOP2 | VLMOP3 | ZDT1 | ZDT2 | ZDT3 | ZDT4 | ZDT6 |
|---|---|---|---|---|---|---|---|---|---|---|---|
| $\mathcal{D}$(best) | 10.43 | 8.32 | 3.87 | 0.08 | 1.64 | 45.14 | 4.04 | 4.70 | 5.05 | 5.46 | 4.76 |
| E2E | 10.06 ± 0.00 | 6.37 ± 0.07 | 4.35 ± 0.00 | 0.00 ± 0.00 | 4.18 ± 0.02 | 46.76 ± 0.09 | 2.69 ± 0.00 | 3.21 ± 0.00 | 5.46 ± 0.00 | 3.04 ± 0.02 | 4.87 ± 0.02 |
| E2E + GradNorm | **10.65 ± 0.00** | 8.62 ± 2.08 | 2.32 ± 0.04 | 0.00 ± 0.00 | 2.67 ± 1.21 | 38.20 ± 0.17 | 4.76 ± 0.00 | 4.01 ± 0.17 | 5.27 ± 0.03 | 3.02 ± 0.02 | 2.55 ± 0.23 |
| E2E + PcGrad | 10.62 ± 0.02 | **10.52 ± 0.00** | 4.32 ± 0.03 | 1.36 ± 1.36 | 4.08 ± 0.10 | 34.65 ± 0.06 | 4.37 ± 0.29 | 5.70 ± 0.01 | 4.45 ± 0.94 | 2.99 ± 0.02 | 1.87 ± 0.10 |
| MH | 10.37 ± 0.24 | **10.61 ± 0.09** | 4.29 ± 0.05 | 0.95 ± 0.95 | 4.18 ± 0.01 | 46.78 ± 0.10 | 2.69 ± 0.00 | 4.48 ± 1.27 | 5.50 ± 0.03 | 2.94 ± 0.08 | **4.90 ± 0.00** |
| MH + GradNorm | 10.64 ± 0.00 | 9.58 ± 0.93 | 3.43 ± 0.90 | 0.00 ± 0.00 | 4.06 ± 0.01 | 29.52 ± 0.54 | 4.82 ± 0.01 | 4.32 ± 0.24 | 4.14 ± 1.07 | 3.16 ± 0.06 | 4.83 ± 0.03 |
| MH + PcGrad | 10.61 ± 0.01 | 10.36 ± 0.02 | 4.34 ± 0.00 | 1.47 ± 1.47 | 2.66 ± 1.21 | 45.33 ± 1.58 | 2.69 ± 0.01 | 5.68 ± 0.04 | 5.38 ± 0.02 | 3.49 ± 0.18 | 2.06 ± 0.15 |
| MM | 10.64 ± 0.00 | **10.56 ± 0.03** | 4.35 ± 0.00 | 0.56 ± 0.56 | **4.22 ± 0.00** | **46.93 ± 0.00** | 4.75 ± 0.00 | 5.56 ± 0.00 | **5.71 ± 0.01** | 3.70 ± 0.38 | 4.87 ± 0.01 |
| MM + COMs | 10.55 ± 0.04 | 8.73 ± 0.01 | 3.85 ± 0.21 | 0.00 ± 0.00 | 1.68 ± 0.01 | 44.05 ± 0.26 | 3.82 ± 0.16 | 4.66 ± 0.11 | 5.44 ± 0.07 | 4.31 ± 0.05 | 4.33 ± 0.01 |
| MM + RoMA | 10.53 ± 0.06 | 10.01 ± 0.08 | 2.60 ± 0.01 | 0.00 ± 0.00 | 1.46 ± 0.00 | 40.48 ± 0.34 | **4.86 ± 0.01** | 5.62 ± 0.01 | 5.40 ± 0.18 | 2.87 ± 0.09 | 1.76 ± 0.02 |
| MM + IOM | 10.61 ± 0.00 | **10.55 ± 0.15** | 4.34 ± 0.00 | 0.58 ± 0.58 | 3.73 ± 0.63 | 46.92 ± 0.00 | 4.62 ± 0.03 | 5.72 ± 0.00 | 5.50 ± 0.01 | 4.39 ± 0.44 | 4.86 ± 0.00 |
| MM + ICT | 10.63 ± 0.00 | 9.94 ± 0.05 | 3.93 ± 0.00 | 0.06 ± 0.06 | 1.46 ± 0.00 | 43.55 ± 2.98 | 3.45 ± 0.07 | 5.50 ± 0.01 | 4.14 ± 0.12 | 3.27 ± 0.09 | 1.88 ± 0.01 |
| MM + Tri-Mentor | 10.61 ± 0.01 | 9.76 ± 0.01 | 3.39 ± 0.00 | **3.73 ± 0.07** | 1.46 ± 0.00 | 46.56 ± 0.08 | 4.33 ± 0.05 | 5.04 ± 0.01 | 5.45 ± 0.04 | 3.21 ± 0.22 | 1.90 ± 0.00 |
| MOEA/D + MM | 10.03 ± 0.01 | 10.36 ± 0.05 | 4.77 ± 0.00 | 0.31 ± 0.02 | 4.01 ± 0.01 | 45.36 ± 0.09 | 4.44 ± 0.04 | 5.29 ± 0.05 | 5.38 ± 0.08 | 4.87 ± 0.15 | 4.78 ± 0.01 |
| MOBO | 10.64 ± 0.00 | 8.00 ± 0.03 | 4.25 ± 0.01 | 0.00 ± 0.00 | 1.46 ± 0.00 | 46.91 ± 0.00 | 4.30 ± 0.01 | 4.34 ± 0.01 | 4.99 ± 0.04 | 3.88 ± 0.00 | 2.63 ± 0.11 |
| MOBO-$q$ParEGO | 10.55 ± 0.08 | 9.85 ± 0.14 | 4.19 ± 0.09 | 0.00 ± 0.00 | 1.46 ± 0.00 | 46.82 ± 0.03 | 4.11 ± 0.08 | 4.66 ± 0.06 | 4.96 ± 0.11 | 4.31 ± 0.07 | 2.51 ± 0.65 |
| MOBO-JES | 10.26 ± 0.10 | 8.75 ± 0.07 | 2.98 ± 0.00 | N/A | 1.46 ± 0.00 | 45.77 ± 0.64 | 3.87 ± 0.04 | 3.90 ± 0.02 | 4.72 ± 0.10 | 3.97 ± 0.24 | 1.87 ± 0.10 |
| PROUD | 10.39 ± 0.06 | 8.85 ± 0.09 | 4.77 ± 0.01 | 2.89 ± 0.28 | 4.00 ± 0.01 | 45.22 ± 0.05 | 4.16 ± 0.09 | 6.00 ± 0.26 | 5.20 ± 0.11 | 4.71 ± 0.05 | 4.29 ± 0.06 |
| LaMBO-2 | 10.39 ± 0.06 | 8.93 ± 0.10 | 4.77 ± 0.01 | 2.78 ± 0.02 | 4.01 ± 0.00 | 40.17 ± 2.33 | 4.17 ± 0.09 | 5.16 ± 0.04 | 5.04 ± 0.14 | 4.74 ± 0.06 | 4.28 ± 0.06 |
| CorrVAE | 10.28 ± 0.10 | 8.83 ± 0.07 | 4.76 ± 0.01 | 2.88 ± 0.05 | 3.99 ± 0.00 | 39.69 ± 1.55 | 4.11 ± 0.05 | 5.82 ± 0.12 | 4.97 ± 0.21 | 4.66 ± 0.02 | 4.28 ± 0.06 |
| MOGFN | 10.33 ± 0.06 | 8.88 ± 0.07 | 4.77 ± 0.01 | 3.01 ± 0.24 | 4.00 ± 0.01 | 44.73 ± 0.14 | 4.14 ± 0.03 | 5.92 ± 0.15 | 5.08 ± 0.02 | 4.68 ± 0.03 | 4.31 ± 0.04 |
| ParetoFlow (**ours**) | 10.58 ± 0.01 | 9.22 ± 0.05 | **4.78 ± 0.00** | 3.01 ± 0.26 | 4.06 ± 0.02 | 46.70 ± 0.03 | 4.29 ± 0.04 | **6.73 ± 0.13** | 5.44 ± 0.11 | **5.06 ± 0.06** | 4.48 ± 0.02 |

Table 16: Hypervolume results for MO-NAS (Part 1).

| Methods | C-10/MOP1 | C-10/MOP2 | C-10/MOP3 | C-10/MOP4 | C-10/MOP5 | C-10/MOP6 | C-10/MOP7 | C-10/MOP8 | C-10/MOP9 |
|---|---|---|---|---|---|---|---|---|---|
| $\mathcal{D}$(best) | 4.72 | 10.42 | 9.21 | 18.62 | 40.79 | 103.55 | 399.67 | 4.38 | 9.64 |
| E2E | 4.69 ± 0.00 | 10.27 ± 0.15 | 9.92 ± 0.05 | **19.97 ± 0.39** | 48.17 ± 0.19 | 102.75 ± 1.38 | 486.43 ± 1.47 | 4.36 ± 0.00 | 9.88 ± 0.03 |
| E2E + GradNorm | 4.58 ± 0.04 | 10.41 ± 0.01 | 9.02 ± 0.08 | 18.06 ± 0.22 | 47.65 ± 0.76 | 90.23 ± 7.18 | 476.94 ± 8.36 | 3.28 ± 0.14 | 8.24 ± 0.05 |
| E2E + PcGrad | 4.68 ± 0.04 | 10.42 ± 0.01 | 9.97 ± 0.06 | **20.12 ± 0.15** | 48.31 ± 0.38 | 104.21 ± 0.08 | **490.41 ± 0.00** | 3.93 ± 0.13 | 9.58 ± 0.36 |
| MH | 4.59 ± 0.04 | 10.41 ± 0.02 | 9.82 ± 0.06 | **20.23 ± 0.32** | **48.61 ± 0.04** | 101.41 ± 2.66 | **490.70 ± 0.88** | 4.43 ± 0.06 | 8.89 ± 0.05 |
| MH + GradNorm | 4.71 ± 0.00 | 10.18 ± 0.02 | 8.48 ± 0.40 | 14.55 ± 2.92 | 38.88 ± 4.82 | 69.78 ± 5.52 | 329.09 ± 10.78 | 3.46 ± 0.05 | 9.20 ± 0.64 |
| MH + PcGrad | 4.71 ± 0.00 | 10.19 ± 0.20 | 9.71 ± 0.11 | 20.20 ± 0.29 | 47.18 ± 0.85 | 101.97 ± 1.71 | 480.97 ± 2.45 | 3.88 ± 0.14 | 9.78 ± 0.03 |
| MM | 4.68 ± 0.01 | 10.37 ± 0.00 | **10.03 ± 0.00** | 19.96 ± 0.29 | 47.33 ± 0.00 | 100.94 ± 0.40 | 484.29 ± 0.00 | 4.33 ± 0.00 | 9.81 ± 0.00 |
| MM + COM | 4.72 ± 0.00 | 10.36 ± 0.00 | 9.89 ± 0.00 | 19.68 ± 0.00 | 45.41 ± 0.00 | 103.52 ± 0.37 | 466.90 ± 0.00 | 4.23 ± 0.05 | 9.57 ± 0.00 |
| MM + IOM | 4.69 ± 0.00 | 10.43 ± 0.00 | 9.96 ± 0.00 | **20.00 ± 0.07** | 45.95 ± 0.00 | 102.60 ± 0.78 | 481.43 ± 0.00 | **4.54 ± 0.00** | **9.89 ± 0.00** |
| MM + RoMA | **4.73 ± 0.00** | 10.42 ± 0.00 | 10.01 ± 0.00 | **20.06 ± 0.05** | 47.64 ± 0.00 | 101.98 ± 1.65 | 472.22 ± 0.00 | 4.15 ± 0.00 | 9.52 ± 0.04 |
| MM + ICT | 4.65 ± 0.04 | 10.13 ± 0.00 | 9.48 ± 0.00 | 19.55 ± 0.00 | 45.80 ± 0.00 | 104.72 ± 0.31 | 471.56 ± 0.00 | 4.28 ± 0.00 | 9.16 ± 0.00 |
| MM + Tri-Mentor | 4.69 ± 0.00 | 10.40 ± 0.00 | 9.96 ± 0.00 | 19.77 ± 0.21 | 38.00 ± 0.00 | 102.07 ± 0.09 | 465.07 ± 4.15 | 4.24 ± 0.02 | 8.22 ± 0.36 |
| MOEA/D + MM | **4.73 ± 0.05** | 9.87 ± 0.01 | 9.33 ± 0.00 | **20.24 ± 0.34** | 47.58 ± 0.05 | 101.33 ± 0.04 | 470.63 ± 3.38 | 4.12 ± 0.04 | 8.93 ± 0.08 |
| MOBO | 4.59 ± 0.00 | 10.36 ± 0.02 | 8.56 ± 0.00 | 18.73 ± 0.48 | 40.27 ± 0.02 | 100.63 ± 2.81 | 482.04 ± 6.29 | 4.14 ± 0.00 | 7.99 ± 0.01 |
| MOBO-$q$ParEGO | 4.53 ± 0.07 | 8.39 ± 0.04 | 8.45 ± 0.19 | 19.62 ± 0.44 | 37.17 ± 0.02 | 91.53 ± 5.93 | 337.57 ± 8.91 | 4.13 ± 0.06 | 8.23 ± 0.15 |
| MOBO-JES | N/A | N/A | N/A | N/A | N/A | N/A | N/A | N/A | N/A |
| PROUD | 4.65 ± 0.07 | 10.40 ± 0.03 | 8.37 ± 0.10 | 17.36 ± 0.56 | 45.04 ± 2.82 | 101.67 ± 2.00 | 445.12 ± 6.42 | 4.06 ± 0.19 | 8.63 ± 0.47 |
| LaMBO-2 | 4.66 ± 0.07 | 10.41 ± 0.03 | 8.37 ± 0.10 | 17.08 ± 0.37 | 44.05 ± 2.23 | 102.18 ± 2.01 | 444.63 ± 7.55 | 4.20 ± 0.03 | 8.74 ± 0.51 |
| CorrVAE | 4.55 ± 0.09 | 10.39 ± 0.03 | 8.12 ± 0.13 | 16.75 ± 0.29 | 44.42 ± 2.32 | 99.74 ± 0.94 | 438.30 ± 7.19 | 3.87 ± 0.10 | 8.13 ± 0.11 |
| MOGFN | 4.60 ± 0.04 | 10.40 ± 0.02 | 8.20 ± 0.14 | 16.92 ± 0.07 | 45.70 ± 0.96 | 100.25 ± 0.81 | 442.90 ± 4.79 | 3.96 ± 0.12 | 8.25 ± 0.22 |
| ParetoFlow (**ours**) | **4.74 ± 0.01** | **10.47 ± 0.01** | 9.32 ± 0.18 | 19.81 ± 0.43 | 47.63 ± 1.00 | **105.76 ± 0.75** | 489.21 ± 3.14 | 4.37 ± 0.04 | 9.66 ± 0.07 |

Table 17: Hypervolume results for MO-NAS (Part 2).

| Methods | IN-1K/MOP1 | IN-1K/MOP2 | IN-1K/MOP3 | IN-1K/MOP4 | IN-1K/MOP5 | IN-1K/MOP6 | IN-1K/MOP7 | IN-1K/MOP8 | IN-1K/MOP9 | NasBench201-Test |
|---|---|---|---|---|---|---|---|---|---|---|
| $\mathcal{D}$(best) | 4.36 | 4.45 | 9.86 | 4.15 | 4.30 | 9.15 | 3.70 | 9.13 | 18.87 | 9.89 |
| E2E | $4.55 \pm 0.03$ | $4.49 \pm 0.02$ | $9.88 \pm 0.12$ | $4.38 \pm 0.03$ | $\mathbf{4.62 \pm 0.08}$ | $9.46 \pm 0.20$ | $3.93 \pm 0.10$ | $9.39 \pm 0.07$ | $19.30 \pm 0.37$ | $8.94 \pm 0.11$ |
| E2E + GradNorm | $4.00 \pm 0.02$ | $4.30 \pm 0.04$ | $7.95 \pm 0.02$ | $4.08 \pm 0.16$ | $3.94 \pm 0.36$ | $7.08 \pm 1.30$ | $3.49 \pm 0.15$ | $8.24 \pm 0.21$ | $16.88 \pm 0.68$ | $8.64 \pm 0.09$ |
| E2E + PcGrad | $4.45 \pm 0.03$ | $4.38 \pm 0.10$ | $\mathbf{9.98 \pm 0.02}$ | $4.15 \pm 0.12$ | $4.40 \pm 0.01$ | $9.43 \pm 0.04$ | $3.64 \pm 0.02$ | $9.29 \pm 0.05$ | $19.37 \pm 0.30$ | $9.03 \pm 0.11$ |
| MH | $4.50 \pm 0.06$ | $4.46 \pm 0.10$ | $9.91 \pm 0.14$ | $4.43 \pm 0.06$ | $\mathbf{4.57 \pm 0.02}$ | $\mathbf{9.66 \pm 0.06}$ | $4.15 \pm 0.13$ | $9.27 \pm 0.03$ | $20.00 \pm 0.13$ | $8.82 \pm 0.11$ |
| MH + GradNorm | $4.15 \pm 0.02$ | $3.68 \pm 0.51$ | $8.75 \pm 1.08$ | $3.89 \pm 0.53$ | $4.38 \pm 0.05$ | $8.97 \pm 0.24$ | $2.62 \pm 0.16$ | $4.71 \pm 1.27$ | $9.43 \pm 1.93$ | $8.56 \pm 0.11$ |
| MH + PcGrad | $4.44 \pm 0.05$ | $\mathbf{4.50 \pm 0.00}$ | $9.95 \pm 0.04$ | $4.15 \pm 0.07$ | $4.36 \pm 0.05$ | $9.34 \pm 0.18$ | $3.86 \pm 0.04$ | $9.33 \pm 0.13$ | $19.31 \pm 0.35$ | $9.07 \pm 0.04$ |
| MM | $4.52 \pm 0.00$ | $4.44 \pm 0.00$ | $9.95 \pm 0.00$ | $\mathbf{4.45 \pm 0.00}$ | $4.42 \pm 0.00$ | $9.25 \pm 0.47$ | $4.00 \pm 0.13$ | $9.43 \pm 0.02$ | $19.66 \pm 0.38$ | $8.94 \pm 0.06$ |
| MM + COMs | $4.17 \pm 0.01$ | $4.21 \pm 0.06$ | $7.54 \pm 0.04$ | $4.23 \pm 0.08$ | $4.33 \pm 0.02$ | $\mathbf{9.51 \pm 0.12}$ | $3.70 \pm 0.14$ | $9.40 \pm 0.04$ | $19.81 \pm 0.13$ | $8.01 \pm 0.49$ |
| MM + RoMA | $\mathbf{4.58 \pm 0.00}$ | $4.54 \pm 0.00$ | $\mathbf{9.97 \pm 0.00}$ | $4.19 \pm 0.05$ | $4.36 \pm 0.00$ | $9.36 \pm 0.15$ | $3.62 \pm 0.01$ | $\mathbf{9.54 \pm 0.03}$ | $\mathbf{20.06 \pm 0.04}$ | $8.92 \pm 0.09$ |
| MM + IOM | $\mathbf{4.58 \pm 0.00}$ | $4.27 \pm 0.03$ | $9.91 \pm 0.05$ | $4.38 \pm 0.06$ | $4.34 \pm 0.02$ | $\mathbf{9.67 \pm 0.07}$ | $\mathbf{4.29 \pm 0.10}$ | $9.34 \pm 0.02$ | $19.49 \pm 0.37$ | $8.70 \pm 0.00$ |
| MM + ICT | $4.49 \pm 0.00$ | $4.20 \pm 0.00$ | $9.81 \pm 0.00$ | $4.30 \pm 0.03$ | $4.31 \pm 0.02$ | $\mathbf{9.62 \pm 0.01}$ | $3.48 \pm 0.07$ | $9.19 \pm 0.37$ | $18.71 \pm 0.84$ | $8.90 \pm 0.14$ |
| MM + Tri-Mentor | $4.17 \pm 0.05$ | $4.26 \pm 0.01$ | $9.75 \pm 0.02$ | $4.14 \pm 0.01$ | $4.28 \pm 0.03$ | $9.40 \pm 0.26$ | $3.97 \pm 0.02$ | $9.13 \pm 0.17$ | $14.81 \pm 1.74$ | $8.75 \pm 0.00$ |
| MOEA/D + MM | $4.13 \pm 0.05$ | $4.46 \pm 0.07$ | $9.67 \pm 0.10$ | $4.27 \pm 0.04$ | $4.37 \pm 0.02$ | $\mathbf{9.72 \pm 0.22}$ | $3.87 \pm 0.09$ | $7.60 \pm 0.14$ | $13.93 \pm 0.56$ | $8.31 \pm 0.07$ |
| MOBO | $4.03 \pm 0.01$ | $4.32 \pm 0.05$ | $7.76 \pm 0.04$ | $4.03 \pm 0.06$ | $4.26 \pm 0.06$ | $8.89 \pm 0.02$ | $3.17 \pm 0.05$ | $8.82 \pm 0.27$ | $15.07 \pm 0.18$ | $8.52 \pm 0.06$ |
| MOBO-$q$ParEGO | $3.62 \pm 0.00$ | $3.97 \pm 0.10$ | $7.95 \pm 0.12$ | $4.00 \pm 0.03$ | $4.06 \pm 0.01$ | $8.93 \pm 0.04$ | $3.81 \pm 0.11$ | $7.99 \pm 0.23$ | $13.85 \pm 0.37$ | $8.68 \pm 0.09$ |
| MOBO-JES | N/A | N/A | N/A | N/A | N/A | N/A | N/A | N/A | N/A | $8.96 \pm 0.16$ |
| PROUD | $4.32 \pm 0.10$ | $4.18 \pm 0.04$ | $9.20 \pm 0.08$ | $3.91 \pm 0.22$ | $3.97 \pm 0.09$ | $9.10 \pm 0.25$ | $3.65 \pm 0.12$ | $7.83 \pm 0.48$ | $16.11 \pm 1.11$ | $\mathbf{9.70 \pm 0.40}$ |
| LaMBO-2 | $4.38 \pm 0.02$ | $4.19 \pm 0.02$ | $9.28 \pm 0.04$ | $3.81 \pm 0.10$ | $3.97 \pm 0.09$ | $8.94 \pm 0.24$ | $3.72 \pm 0.04$ | $7.64 \pm 0.46$ | $16.43 \pm 1.26$ | $\mathbf{9.68 \pm 0.40}$ |
| CorrVAE | $4.25 \pm 0.08$ | $4.16 \pm 0.04$ | $9.13 \pm 0.09$ | $3.73 \pm 0.04$ | $3.97 \pm 0.09$ | $8.82 \pm 0.18$ | $3.51 \pm 0.07$ | $7.44 \pm 0.17$ | $14.48 \pm 0.49$ | $9.57 \pm 0.30$ |
| MOGFN | $4.29 \pm 0.06$ | $4.18 \pm 0.03$ | $9.19 \pm 0.06$ | $3.76 \pm 0.03$ | $4.01 \pm 0.09$ | $8.93 \pm 0.13$ | $3.57 \pm 0.08$ | $7.54 \pm 0.16$ | $15.02 \pm 0.65$ | $\mathbf{9.74 \pm 0.08}$ |
| ParetoFlow (**ours**) | $4.33 \pm 0.01$ | $4.37 \pm 0.06$ | $9.82 \pm 0.08$ | $4.21 \pm 0.05$ | $\mathbf{4.62 \pm 0.05}$ | $9.29 \pm 0.00$ | $3.74 \pm 0.10$ | $9.18 \pm 0.14$ | $18.71 \pm 0.39$ | $9.13 \pm 0.00$ |

Table 18: Hypervolume results for MORL.

| Methods | MO-Hopper | MO-Swimmer |
|---|---|---|
| $\mathcal{D}$(best) | 4.21 | 2.85 |
| E2E | $3.68 \pm 0.00$ | $2.04 \pm 0.10$ |
| E2E + GradNorm | $3.94 \pm 0.23$ | $2.08 \pm 0.02$ |
| E2E + PcGrad | $3.72 \pm 0.01$ | $1.90 \pm 0.05$ |
| MH | $3.74 \pm 0.07$ | $2.66 \pm 0.04$ |
| MH + GradNorm | $3.67 \pm 0.00$ | $1.98 \pm 0.12$ |
| MH + PcGrad | $3.86 \pm 0.18$ | $2.08 \pm 0.02$ |
| MM | $3.76 \pm 0.01$ | $1.91 \pm 0.02$ |
| MM + COMs | $3.72 \pm 0.02$ | $1.98 \pm 0.01$ |
| MM + RoMA | $4.74 \pm 0.00$ | $1.95 \pm 0.06$ |
| MM + IOM | $4.17 \pm 0.18$ | $1.96 \pm 0.06$ |
| MM + ICT | $3.70 \pm 0.01$ | $2.38 \pm 0.11$ |
| MM + Tri-Mentor | $3.82 \pm 0.03$ | $1.98 \pm 0.01$ |
| MOEA/D + MM | $4.75 \pm 0.28$ | $0.86 \pm 0.19$ |
| MOBO | $3.68 \pm 0.00$ | $1.49 \pm 0.02$ |
| MOBO-$q$ParEGO | N/A | N/A |
| MOBO-JES | N/A | N/A |
| PROUD | $4.84 \pm 0.14$ | $2.32 \pm 0.24$ |
| LaMBO-2 | $4.77 \pm 0.00$ | $2.41 \pm 0.24$ |
| CorrVAE | $4.76 \pm 0.01$ | $2.23 \pm 0.20$ |
| MOGFN | $4.78 \pm 0.03$ | $2.35 \pm 0.21$ |
| ParetoFlow (**ours**) | $\mathbf{5.56 \pm 0.01}$ | $\mathbf{2.95 \pm 0.09}$ |

Table 19: Hypervolume results for scientific design.

| Methods | Molecule | Regex | RFP | ZINC |
|---|---|---|---|---|
| $\mathcal{D}$(best) | 2.26 | 3.05 | 3.75 | 4.06 |
| E2E | $1.07 \pm 0.07$ | $2.05 \pm 0.00$ | $3.64 \pm 0.05$ | $3.95 \pm 0.04$ |
| E2E + GradNorm | $1.07 \pm 0.07$ | $2.05 \pm 0.00$ | $3.73 \pm 0.04$ | $3.92 \pm 0.00$ |
| E2E + PcGrad | $\mathbf{2.12 \pm 0.04}$ | $2.05 \pm 0.00$ | $3.70 \pm 0.05$ | $3.89 \pm 0.06$ |
| MH | $\mathbf{2.08 \pm 0.00}$ | $2.05 \pm 0.00$ | $3.74 \pm 0.00$ | $3.86 \pm 0.02$ |
| MH + GradNorm | $1.00 \pm 0.00$ | $2.05 \pm 0.00$ | $3.69 \pm 0.01$ | $3.82 \pm 0.01$ |
| MH + PcGrad | $1.00 \pm 0.00$ | $2.05 \pm 0.00$ | $3.68 \pm 0.02$ | $3.86 \pm 0.01$ |
| MM | $1.10 \pm 0.09$ | $2.05 \pm 0.00$ | $3.70 \pm 0.01$ | $3.84 \pm 0.00$ |
| MM + COMs | $1.76 \pm 0.14$ | $2.38 \pm 0.33$ | $3.70 \pm 0.00$ | $3.86 \pm 0.02$ |
| MM + RoMA | $1.03 \pm 0.00$ | $2.05 \pm 0.00$ | $3.79 \pm 0.04$ | $3.91 \pm 0.02$ |
| MM + IOM | $1.02 \pm 0.01$ | $2.05 \pm 0.00$ | $3.76 \pm 0.03$ | $3.91 \pm 0.02$ |
| MM + ICT | $1.02 \pm 0.02$ | $2.05 \pm 0.00$ | $3.67 \pm 0.00$ | $3.96 \pm 0.07$ |
| MM + Tri-Mentor | $1.41 \pm 0.17$ | $2.05 \pm 0.00$ | $3.75 \pm 0.03$ | $3.75 \pm 0.00$ |
| MOEA/D + MM | $1.47 \pm 0.09$ | $2.99 \pm 0.00$ | $3.62 \pm 0.33$ | $4.52 \pm 0.05$ |
| MOBO | $1.02 \pm 0.02$ | $\mathbf{3.42 \pm 0.25}$ | $3.70 \pm 0.01$ | $3.90 \pm 0.01$ |
| MOBO-$q$ParEGO | $1.96 \pm 0.12$ | $3.17 \pm 0.11$ | $3.33 \pm 0.04$ | $4.00 \pm 0.03$ |
| MOBO-JES | $1.00 \pm 0.00$ | N/A | N/A | N/A |
| PROUD | $1.67 \pm 0.16$ | $\mathbf{3.26 \pm 0.00}$ | $4.15 \pm 0.14$ | $4.26 \pm 0.22$ |
| LaMBO-2 | $1.67 \pm 0.16$ | $\mathbf{3.26 \pm 0.00}$ | $4.09 \pm 0.18$ | $4.17 \pm 0.28$ |
| CorrVAE | $1.58 \pm 0.07$ | $\mathbf{3.26 \pm 0.00}$ | $4.07 \pm 0.07$ | $4.09 \pm 0.18$ |
| MOGFN | $1.60 \pm 0.03$ | $\mathbf{3.26 \pm 0.00}$ | $\mathbf{4.30 \pm 0.06}$ | $4.19 \pm 0.14$ |
| ParetoFlow (**ours**) | $1.99 \pm 0.09$ | $\mathbf{3.26 \pm 0.00}$ | $4.18 \pm 0.04$ | $\mathbf{4.43 \pm 0.04}$ |

Table 20: Hypervolume results for RE.

| Methods | RE21 | RE22 | RE23 | RE24 | RE25 | RE31 | RE32 | RE33 | RE34 | RE35 | RE36 | RE37 | RE41 | RE42 | RE61 | MO-Portfolio |
|---|---|---|---|---|---|---|---|---|---|---|---|---|---|---|---|---|
| $\mathcal{D}$(best) | 4.10 | 4.78 | 4.75 | 4.59 | 4.79 | 10.23 | 10.53 | 10.59 | 9.30 | 10.08 | 7.61 | 4.72 | 18.27 | 14.52 | 97.49 | 3.78 |
| E2E | **4.59 ± 0.00** | 4.84 ± 0.00 | 4.84 ± 0.00 | 4.38 ± 0.00 | 4.73 ± 0.04 | 10.56 ± 0.00 | 10.64 ± 0.00 | 10.68 ± 0.00 | 10.07 ± 0.03 | 9.99 ± 0.52 | **9.92 ± 0.20** | 4.67 ± 0.35 | 19.85 ± 0.27 | 21.06 ± 1.47 | 108.78 ± 0.13 | 2.97 ± 0.14 |
| E2E + GradNorm | 4.54 ± 0.02 | 4.84 ± 0.00 | 2.64 ± 0.00 | 4.29 ± 0.00 | 4.84 ± 0.00 | **10.65 ± 0.00** | 10.61 ± 0.00 | 9.72 ± 0.03 | 8.86 ± 0.75 | 10.35 ± 0.00 | 3.59 ± 2.77 | 6.02 ± 0.07 | 19.46 ± 0.10 | 17.52 ± 0.82 | 108.55 ± 0.31 | 3.14 ± 0.14 |
| E2E + PcGrad | **4.59 ± 0.00** | 4.52 ± 0.32 | 4.84 ± 0.00 | 4.22 ± 0.02 | 4.35 ± 0.00 | **10.65 ± 0.00** | 10.64 ± 0.00 | 9.86 ± 0.36 | 10.04 ± 0.03 | 10.52 ± 0.07 | 9.32 ± 0.07 | 4.00 ± 0.18 | 20.38 ± 0.19 | 21.85 ± 0.53 | 108.57 ± 0.04 | 1.99 ± 0.27 |
| MH | **4.59 ± 0.01** | 4.83 ± 0.01 | 4.59 ± 0.10 | 4.11 ± 0.01 | 3.82 ± 0.30 | 10.64 ± 0.00 | 10.64 ± 0.00 | 10.47 ± 0.22 | 10.02 ± 0.03 | 10.41 ± 0.12 | **9.77 ± 0.31** | 4.43 ± 0.01 | 20.39 ± 0.12 | 21.23 ± 1.80 | 108.87 ± 0.00 | 2.02 ± 0.22 |
| MH + GradNorm | 4.03 ± 0.53 | 3.75 ± 0.06 | 3.70 ± 0.09 | 2.64 ± 0.00 | 3.14 ± 0.01 | **10.65 ± 0.00** | 10.62 ± 0.01 | 6.12 ± 0.49 | 9.65 ± 0.28 | 10.18 ± 0.41 | 6.67 ± 2.32 | 5.90 ± 0.44 | 17.98 ± 3.31 | 14.49 ± 6.08 | 108.17 ± 0.36 | 3.06 ± 0.09 |
| MH + PcGrad | 4.51 ± 0.09 | 4.84 ± 0.00 | 3.42 ± 0.57 | 3.77 ± 0.00 | 4.35 ± 0.00 | 7.64 ± 0.00 | 10.08 ± 0.00 | 10.11 ± 0.35 | 10.04 ± 0.03 | 10.48 ± 0.08 | 9.16 ± 0.26 | 6.32 ± 0.05 | 20.41 ± 0.08 | 21.77 ± 0.73 | 108.39 ± 0.69 | 3.00 ± 0.05 |
| MM | **4.58 ± 0.00** | 4.84 ± 0.00 | 4.84 ± 0.00 | 4.79 ± 0.01 | 4.83 ± 0.01 | 10.63 ± 0.00 | 10.63 ± 0.00 | 9.62 ± 0.62 | 10.07 ± 0.01 | 10.56 ± 0.01 | **9.77 ± 0.04** | 6.45 ± 0.01 | 20.42 ± 0.11 | 22.48 ± 0.02 | 108.54 ± 0.11 | 3.66 ± 0.01 |
| MM + COMs | 4.30 ± 0.04 | 4.83 ± 0.00 | 4.76 ± 0.02 | 4.59 ± 0.00 | 4.84 ± 0.00 | 5.28 ± 5.28 | 10.62 ± 0.00 | 10.26 ± 0.31 | 9.89 ± 0.00 | 10.24 ± 0.26 | 8.90 ± 0.01 | 5.68 ± 0.20 | 19.74 ± 0.00 | 16.23 ± 0.07 | 104.81 ± 0.00 | 2.10 ± 0.08 |
| MM + RoMA | 4.55 ± 0.00 | 4.84 ± 0.00 | 4.83 ± 0.00 | 3.66 ± 0.01 | 3.40 ± 0.01 | 10.60 ± 0.00 | 10.64 ± 0.00 | 10.11 ± 0.05 | 9.07 ± 0.04 | 10.52 ± 0.03 | 7.52 ± 0.51 | 6.37 ± 0.04 | 20.12 ± 0.03 | 19.14 ± 0.05 | 107.51 ± 0.04 | 2.88 ± 0.03 |
| MM + IOM | **4.58 ± 0.00** | 4.84 ± 0.00 | 4.81 ± 0.02 | 4.28 ± 0.01 | 4.14 ± 0.01 | **10.65 ± 0.00** | 10.65 ± 0.00 | 10.64 ± 0.03 | 9.99 ± 0.03 | 10.55 ± 0.01 | 8.92 ± 0.29 | 6.33 ± 0.08 | 20.29 ± 0.09 | 21.78 ± 0.45 | 107.32 ± 0.27 | 2.88 ± 0.02 |
| MM + ICT | **4.59 ± 0.00** | 4.84 ± 0.00 | 2.76 ± 0.00 | 3.23 ± 0.00 | 4.74 ± 0.00 | 10.62 ± 0.01 | 2.77 ± 0.00 | 9.80 ± 0.50 | 10.05 ± 0.01 | 10.49 ± 0.03 | 9.49 ± 0.07 | 6.14 ± 0.09 | 20.09 ± 0.23 | 21.42 ± 0.52 | 107.30 ± 0.95 | 1.75 ± 0.30 |
| MM + Tri-Mentor | **4.58 ± 0.00** | 4.84 ± 0.00 | 2.76 ± 0.00 | 4.81 ± 0.01 | 4.70 ± 0.00 | **10.65 ± 0.00** | 10.65 ± 0.00 | 10.54 ± 0.00 | 10.03 ± 0.04 | 10.57 ± 0.01 | 6.43 ± 0.12 | 6.35 ± 0.07 | 20.37 ± 0.07 | 21.05 ± 0.57 | 107.12 ± 1.06 | 2.50 ± 0.08 |
| MOEA/D + MM | 4.31 ± 0.04 | 4.84 ± 0.00 | 4.84 ± 0.02 | 4.81 ± 0.05 | 4.35 ± 0.13 | 10.31 ± 0.02 | 10.49 ± 0.03 | 10.48 ± 0.02 | 9.56 ± 0.06 | 10.40 ± 0.02 | **9.79 ± 0.21** | 6.60 ± 0.07 | **20.99 ± 0.28** | 21.00 ± 0.18 | 107.73 ± 0.25 | 3.18 ± 0.22 |
| MOBO | 4.31 ± 0.05 | 4.84 ± 0.00 | 4.18 ± 0.01 | 3.32 ± 0.02 | 4.83 ± 0.00 | 10.03 ± 0.00 | 10.53 ± 0.12 | 10.48 ± 0.02 | 9.82 ± 0.35 | 9.42 ± 0.07 | 0.00 ± 0.00 | 6.40 ± 0.08 | 19.27 ± 0.06 | 12.08 ± 0.00 | N/A | 2.89 ± 0.01 |
| MOBO-$q$ParEGO | 4.07 ± 0.15 | 4.21 ± 0.40 | 4.75 ± 0.01 | 0.00 ± 0.00 | 4.12 ± 0.29 | 5.31 ± 5.31 | 8.82 ± 0.37 | 10.46 ± 0.09 | 8.89 ± 0.32 | 0.00 ± 0.00 | 0.00 ± 0.00 | 5.52 ± 0.04 | N/A | N/A | N/A | 2.90 ± 0.06 |
| MOBO-JES | 3.89 ± 0.03 | 4.57 ± 0.03 | 4.66 ± 0.05 | 4.54 ± 0.00 | 4.80 ± 0.00 | 10.01 ± 0.01 | 10.63 ± 0.01 | 10.52 ± 0.03 | 9.03 ± 0.00 | 10.15 ± 0.04 | 6.46 ± 0.34 | 5.24 ± 0.17 | N/A | N/A | N/A | 3.15 ± 0.21 |
| PROUD | 4.41 ± 0.08 | 4.54 ± 0.05 | 4.70 ± 0.04 | 4.83 ± 0.12 | 4.97 ± 0.17 | 10.09 ± 0.27 | 18.01 ± 5.33 | 10.80 ± 0.57 | 10.79 ± 0.55 | 11.72 ± 0.15 | 7.53 ± 0.32 | 6.86 ± 1.00 | 18.32 ± 0.26 | 34.97 ± 10.29 | 113.65 ± 1.79 | 4.15 ± 0.12 |
| LaMBO-2 | 4.41 ± 0.08 | 4.50 ± 0.04 | 4.68 ± 0.03 | 4.85 ± 0.12 | 4.85 ± 0.16 | 10.06 ± 0.34 | 16.84 ± 6.07 | 10.54 ± 0.10 | 10.70 ± 0.58 | 11.61 ± 0.01 | 7.64 ± 0.22 | 7.05 ± 0.52 | 18.22 ± 0.15 | 33.18 ± 7.22 | 114.17 ± 1.63 | 4.14 ± 0.09 |
| CorrVAE | 4.35 ± 0.05 | 4.50 ± 0.03 | 4.69 ± 0.03 | 4.68 ± 0.06 | 4.94 ± 0.14 | 10.03 ± 0.23 | 14.21 ± 2.37 | 10.46 ± 0.11 | 10.54 ± 0.23 | 11.54 ± 0.21 | 7.34 ± 0.21 | 5.94 ± 0.52 | 18.14 ± 0.20 | 16.34 ± 4.77 | 110.80 ± 3.17 | 4.07 ± 0.04 |
| MOGFN | 4.37 ± 0.04 | 4.53 ± 0.04 | 4.70 ± 0.03 | 4.72 ± 0.07 | 5.04 ± 0.10 | 10.17 ± 0.17 | 15.72 ± 2.66 | 10.54 ± 0.13 | 10.69 ± 0.20 | 11.69 ± 0.11 | 7.46 ± 0.21 | 6.27 ± 0.38 | 18.32 ± 0.26 | 30.14 ± 4.47 | 112.71 ± 1.36 | 4.09 ± 0.01 |
| ParetoFlow (ours) | 4.52 ± 0.05 | **4.97 ± 0.09** | **5.82 ± 0.36** | **5.45 ± 0.06** | **6.17 ± 0.41** | 10.37 ± 0.07 | **32.11 ± 6.21** | **11.94 ± 0.48** | **13.26 ± 0.31** | **12.24 ± 0.15** | 8.58 ± 0.18 | **8.13 ± 0.40** | 20.30 ± 0.49 | **41.49 ± 4.97** | **115.94 ± 0.57** | **4.31 ± 0.03** |

## A.8 VISUALIZATIONS AND CASE STUDY DETAILS

We provide C-10/MOP1 and MO-Hopper visualization results in Figure 11. This features comparisons between offline samples and samples generated by ParetoFlow, clearly demonstrating the superior quality of the latter.

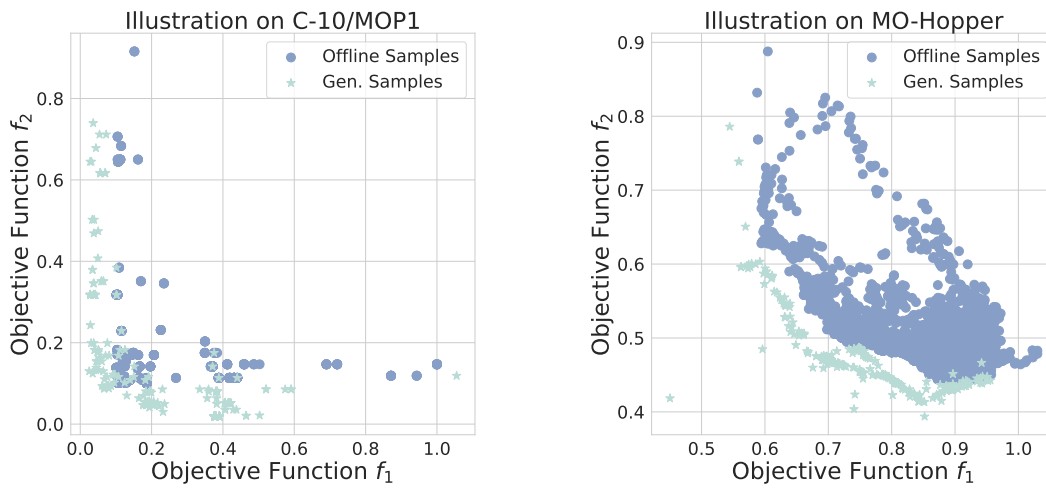

Figure 11: Illustrations of ParetoFlow on two tasks C-10/MOP1 and MO-Hopper.

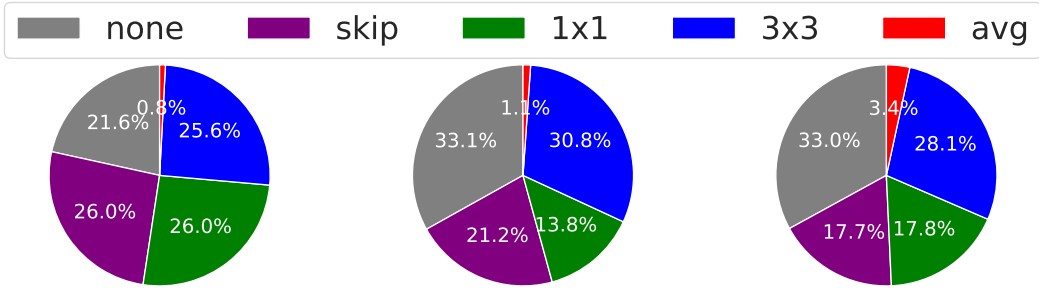

Figure 12: C-10/MOP5 case study: (1) samples prioritizing prediction error and model complexity; (2) samples focusing on prediction error and hardware efficiency; (3) samples emphasizing prediction error, model complexity, and hardware efficiency.

We have conducted a detailed case study on C-10/MOP5, focusing on optimizing prediction error, model complexity, and hardware efficiency, as outlined in Lu et al. (2023). The case study analyzes

three sets of solutions as detailed in Figure 12 where the weight vector for unconsidered objectives is set to zero. We discuss the frequency of operators used in these sets, noting that the 3x3 convolution is consistently preferred across three sets for its effectiveness in reducing prediction error, while the 3x3 average pooling is less favored. Additionally, there is a notable shift from the use of 1x1 convolutions to 'none' operators in moving from the first to the second set, suggesting a trade-off for better hardware efficiency. This analysis provides insights into the structural preferences and performance trade-offs in the generated architectures.

### A.9 Effectiveness of local filtering

To further verify the effectiveness of local filtering, we conduct experiments on the convex-PF task ZDT1 and the nonconvex-PF task ZDT2. When we remove the local filtering, the performance of ZDT1 nearly does not change: from $4.30 \pm 0.02$ to $4.29 \pm 0.04$. In contrast, the performance on ZDT2 drops obviously: from $6.79 \pm 0.16$ to $5.78 \pm 0.15$. This demonstrates the effectiveness of our local filtering scheme in handling nonconvex PFs and verifies its underlying motivation.

### A.10 Further Ablations

To substantiate the advantages of flow matching over diffusion models, we replace flow matching in our ParetoFlow framework with a diffusion model Song et al. (2021) and conduct comparisons on two tasks: MO-Hopper and C-10/MOP1. The results in Table 21 consistently demonstrate the superior performance of flow matching in our context.

Table 21: Comparison between flow matching and diffusion models

| Methods | C-10/MOP1 | MO-Hopper |
| --- | --- | --- |
| ParetoFlow w/ Diffusion | $4.65 \pm 0.05$ | $5.54 \pm 0.07$ |
| ParetoFlow (ours) | $\mathbf{4.77 \pm 0.00}$ | $\mathbf{5.69 \pm 0.03}$ |

We further compare our Das-Deniss with another weight generation strategies. The Das-Dennis method is widely used in multi-objective optimization studies due to its simplicity and ease of use, as it does not require optimization. However, a limitation of this method is that it does not allow for specifying an exact number of weights. On the other hand, the Riesz s-Energy method Hardin & Saff (2005) allows for precise control over the number of weights generated. However, this approach involves a optimization process, making it more complex to implement. We conduct experiments on C-10/MOP1 and MO-Hopper using both strategies, and find that their performance quite close, as shown in Table 22.

Table 22: Comparison between Das-Dennis and Riesz s-Energy

| Methods | C-10/MOP1 | MO-Hopper |
| --- | --- | --- |
| ParetoFlow w/ Riesz s-Energy | $\mathbf{4.77 \pm 0.00}$ | $5.55 \pm 0.10$ |
| ParetoFlow (ours) | $\mathbf{4.77 \pm 0.00}$ | $\mathbf{5.69 \pm 0.03}$ |

### A.11 Sensitivity to the Number of Sampling Steps

As shown in Figure 10, our method is robust to changes in the number of sampling steps $T$.

### A.12 Relationship Between Evolutionary Algorithms and Flow Matching

We further discuss the relationship between evolutionary algorithms(EA) and flow matching. In our flow sampling process, each intermediate noisy sample $x_t$ can be mapped to a clean sample $\hat{x}_1(x_t)$. This mapping bridges flow matching and EA, where flow matching handles $x_t$ and EA operates on $x_1$. Specifically, in our algorithm, we use the weighted predictor $f_\omega(\hat{x}_1(x_t))$ to select promising $x_t$

for the next iteration. This predictor selection, originally part of EA and applied to $x_1$, is integrated into flow matching through its application to $x_t$. This relationship demonstrates the integration of EA and flow matching.

To illustrate the efficacy of integrating EA with flow matching, consider the performance of each component in isolation. Removing flow matching from our framework leaves us solely with EA. Table 1 demonstrates that the EA method NSGA-II, performs worse than our integrated approach, highlighting the value added by flow matching. Conversely, excluding EA results in a system where flow matching operates on uniformly weighted objectives for guided sampling, but lacks the crucial selection and local filtering processes. Our experiments on tasks like MO-Hopper and C-10/MOP1 show that this configuration leads to inferior hypervolume (HV) results, as depicted in Table 23, further validating the significance of combining both strategies:

Table 23: ParetoFlow w/o EA

| Methods | C-10/MOP1 | MO-Hopper |
|---|---|---|
| ParetoFlow w/o EA | $4.53 \pm 0.00$ | $5.06 \pm 0.00$ |
| ParetoFlow (ours) | $\mathbf{4.77 \pm 0.00}$ | $\mathbf{5.69 \pm 0.03}$ |

This comparative analysis clearly supports the effectiveness of our integrated method, demonstrating that each component contributes significantly to the overall performance.

### A.13 ETHICS STATEMENT AND LIMITATIONS

**Ethics Statement.** Our method, *ParetoFlow*, holds promise for accelerating advancements in new materials, biomedical developments, and robotic technologies by simultaneously optimizing multiple desired properties. Such advancements could drive significant progress in these fields. However, like any powerful tool, *ParetoFlow* also carries risks of misuse. A potential concern is the application of this technology in designing systems or devices for malevolent purposes. For example, inappropriately used, the optimization capabilities could aid in developing more effective and energy-efficient robotic weaponry. It is, therefore, imperative to establish robust safeguards and strict regulations to control the application of such technologies, especially in critical sectors.

**Limitation.** While our method shows considerable promise, its effectiveness heavily relies on the accuracy of the underlying predictive models. In highly complex applications such as protein sequence design Ferruz et al. (2022); Chen et al. (2023b; 2022), where amino acid interactions are intricately linked, simple predictive models may fall short in capturing these complexities, leading to suboptimal performance. Consequently, task-specific strategies may be essential for accurately modeling such complex scenarios. For instance, employing advanced protein models Lin et al. (2023); Chen et al. (2023c) could enhance the modeling of protein sequences. Future research should consider integrating domain-specific insights into the predictor modeling, thus improving the method's ability to handle complex challenges more effectively.

