# OpenReview forum: "ParetoFlow: Guided Flows in Multi-Objective Optimization"
_ICLR.cc/2025/Conference — ICLR 2025 Poster_

### Official Review · Reviewer_FugU · 2024-11-02

**Soundness:** 2
**Presentation:** 2
**Contribution:** 3
**Rating:** 6
**Confidence:** 3

**Summary:**

This paper applies flow matching to offline multi-objective optimization. Authors propose a multi-objective predictor guidance module which generates weigts for each objective using Das-Dennis approach to guide the sample generation towards the Pareto front, and neighboring evolution module is proposed to enhance knowledge sharing between the generation of different weights. Experimental results show the effectiveness of the proposed method.

**Strengths:**

1. The concept of ParetoFlow is innovative and creative.

2. The experimental results show that the proposed method has promising performance on the tested problems and surpass the baselines.

**Weaknesses:**

1. The proposed method relies on the accuracy of the objective predictors, which might be a bottleneck on problems with complex landscapes and limited offline dataset.

2. It seems that the required weight vectors will grow exponentially (i.e., a problem with 10 objectives neesds at least 1024 vectors to ensure each objective has at least 2 weights). It may be time consuming to generate and sample using these weights, while less weights may not fill the PF.

**Questions:**

1. In Section 3.1 authors consider the case of non convex PF. I wonder whether such cases show up in the experiemnts? If so, does the designed filtering successfully process such cases?

2. In the Neighboring Update, the next state is the best one in the K nearest weight vectors, will it make a large ratio of vectors generate the same solution and lead to early convergence?

---

> ### Author Response · Authors · 2024-11-16
>
> Dear Reviewer,
>
> Thank you for your invaluable feedback, which has greatly guided our revisions. We have incorporated your suggestions into the manuscript, with all changes clearly marked in red for easier reference.
>
> ## Weaknesses
>
> > The proposed method relies on the accuracy of the objective predictors, which might be a bottleneck on problems with complex landscapes and limited offline dataset.
>
> We acknowledge that the accuracy of the objective predictors is indeed a critical factor, and it can be a limitation, especially for complex problem landscapes and limited offline data.  This challenge is not unique to our method but is a common aspect of all offline optimization frameworks. Nevertheless, our evaluation results in the paper show that our ParetoFlow framework can achieve strong performance in the tested scenarios.
>
> To mitigate this limitation in more complex tasks, one potential future direction is to pre-train a feature extraction model that provides robust representations as input for the predictor. For example, in the protein domain, pre-trained models such as protein BERT have demonstrated strong downstream prediction performance even with limited labeled data. Integrating such techniques could enhance the robustness of the predictors and improve overall performance. We plan to explore this approach and other enhancements in future work.
>
> > It seems that the required weight vectors will grow exponentially (i.e., a problem with 10 objectives neesds at least 1024 vectors to ensure each objective has at least 2 weights). It may be time consuming to generate and sample using these weights, while less weights may not fill the PF.
>
> Generating these weights is straightforward, and we can efficiently generate them using the Das-Dennis method.
>
> While optimizing numerous weighted objectives can indeed be challenging, our method is designed to scale effectively. For instance, if we have 1024 weight vectors, our parallelized code processes a batch of 1024 samples, which can be handled efficiently and in parallel using GPU resources. It is quite common to use batch sizes of 1024 or even up to 10240 in modern computational settings.
>
> Additionally, we would like to note that most real-world problems typically involve only 2 or 3 objectives. Therefore, our framework remains efficient and effective for these more common cases.
>
> ## Questions
>
> > In Section 3.1 authors consider the case of non convex PF. I wonder whether such cases show up in the experiemnts? If so, does the designed filtering successfully process such cases?
>
> Yes, we have already discussed the case of nonconvex Pareto fronts (PFs) in our "Effectiveness of Local Filtering" Appendix A.9. We conduct experiments on the convex-PF task  ZDT$1$ and the nonconvex-PF task ZDT$2$. When we remove the local filtering, the performance of ZDT$1$ nearly does not change: from $4.88 \pm 0.02$ to $4.81 \pm 0.08$. In contrast, the performance on ZDT$2$ drops obviously: from $6.79 \pm 0.16$ to $5.78 \pm 0.15$. This demonstrates the effectiveness of our local filtering scheme in handling nonconvex PFs and verifies its underlying motivation.
>
> > In the Neighboring Update, the next state is the best one in the K nearest weight vectors, will it make a large ratio of vectors generate the same solution and lead to early convergence?
>
> No, early convergence is not an issue as different weighted objectives typically favor different samples. In our analysis, we observed that only 12\% of the points in C-10/MOP1 and 1\% in MO-Hopper were duplicates. The higher duplication rate in C-10/MOP1 is primarily due to the decoding of continuous logits back to the same discrete values. We have detailed this phenomenon and its implications in Section 4.5, Ablation Studies.
>
> ## Overall
>
> We are grateful for your careful review and valuable feedback. Does our response address your concerns? We look forward to further interaction during the rebuttal phase.

---

> > ### Author Response · Authors · 2024-11-21
> > **Looking Forward to Your Feedback**
> >
> > Dear Reviewer FugU,
> >
> > Thank you for your thoughtful review and constructive feedback. We have carefully addressed your concerns as follows:
> >
> > - We have clarified that our evaluation results demonstrate ParetoFlow's ability to achieve strong performance with the current predictors, and we have discussed future directions to address potential inaccuracies in predictors.
> >
> > - We have demonstrated the efficiency of generating weights and sampling using these weights.
> >
> > - We have already discussed the case of nonconvex Pareto fronts (PFs) in Appendix A.9, showcasing the effectiveness of our local filtering approach.
> > - We have clarified that only a small portion of samples are duplicated, which does not lead to early convergence.
> >
> > We sincerely appreciate your valuable insights and welcome any further questions or points of clarification. As the weekend approaches, we are eager to continue this discussion and address any additional feedback. We look forward to hearing from you.

---

> > > ### Comment · Reviewer_FugU · 2024-11-26
> > >
> > > Thanks for your detailed response. Thus I increased my score by one point.

---

> > > > ### Author Response · Authors · 2024-11-26
> > > > **Thank you for your support!**
> > > >
> > > > Thank you for your kind feedback! We sincerely appreciate your recognition and the score adjustment.

---

### Official Review · Reviewer_Cej3 · 2024-11-03

**Soundness:** 3
**Presentation:** 3
**Contribution:** 3
**Rating:** 6
**Confidence:** 4

**Summary:**

This paper propose a new MOO method.

**Strengths:**

1. **The title is engaging and draws interest.**

2. **Overall, the paper is well-written and easy to follow.**

3. **Section 2.2 is clearly explained. However, could you elaborate on how using a normalizing flow would alter or improve the results?**

**Weaknesses:**

1. **Insufficient Mathematical Rigor**: The paper lacks adequate mathematical depth and rigor. Incorporating more detailed mathematical formulations, proofs, and theoretical underpinnings would strengthen the credibility and scholarly value of the work.

2. **Incorrect Formalization in Line 105**: The statement on line 105 is inaccurate. A Multi-Objective Optimization (MOO) problem cannot be "formally" formulated merely as a vector optimization problem. Such a formulation is more of an informal notational convenience rather than a formal definition. Clarification and correct formalism are needed here.

3. **Unclear Purpose of Equation (9)**: The role and application of Equation (9) are not clearly explained. It is difficult to understand its significance and how it fits into the overall argument of the paper. Providing a thorough explanation or context for this equation would enhance comprehension.

4. **Unconvincing Integration of EA and Flow Matching**: The combination of Evolutionary Algorithms (EA) and flow matching presented in the paper does not seem convincing. Additional justification, evidence, or examples demonstrating the effectiveness of this integration would help in substantiating this approach.

5. **Inappropriateness of Section 4.6 in Main Text**: Section 4.6 may not be suitable for inclusion in the main body of the paper. It might be more appropriate to place this section in an appendix or supplementary material to maintain the focus and flow of the main content.

6. **Unconvincing Experimental Results Using Hypervolume Metric**: The experiments conducted are not entirely convincing, particularly due to the exclusive use of the hypervolume (HV) metric. If alternative methods such as SMS-MOEA were employed, it is questionable whether the proposed method would perform as well as, or better than, SMS-MOEA. Including comparisons with such methods and using a variety of performance metrics could provide a more comprehensive evaluation.

**Questions:**

see advantages and weakness.

---

> ### Author Response · Authors · 2024-11-16
>
> Dear Reviewer,
>
> Thank you for your constructive feedback and positive comments on our title, 'ParetoFlow.' Your insights have greatly contributed to improving our paper. We have carefully revised the manuscript to incorporate your suggestions, with all changes highlighted in red for your convenience.
>
> ## Strengths:
>
> > Section 2.2 is clearly explained. However, could you elaborate on how using a normalizing flow would alter or improve the results?
>
> We have now included two additional high-impact papers [r1, r2] in our revised submission to provide stronger evidence that flow matching outperforms diffusion models. Specifically, Section 5.5 (Ablation on generative modeling approaches) in [r1] and Section 3.6.2 (Ablations) in [r2] address this topic in detail.
>
> Furthermore, to empirically substantiate our method's efficacy, we implemented a variant of our ParetoFlow framework using a diffusion model and conducted comparative experiments on two tasks, MO-Hopper and C-10/MOP1, using five random seeds. The results, detailed in the table below and added to our Appendix A.10 titled "Further Ablations", clearly demonstrate the superior performance of flow matching:
>
>
> | Methods                 | C-10/MOP1     | MO-Hopper     |
> | ----------------------- | ------------- | ------------- |
> | ParetoFlow w/ Diffusion | $4.65 ± 0.05$ | $5.54 ± 0.07$ |
> | ParetoFlow (ours)       | $4.77 ± 0.00$ | $5.69 ± 0.03$ |
>
> These empirical findings corroborate our choice of flow matching over diffusion models, supporting our claim with both qualitative and quantitative evidence.
>
>
>     [r1] Le M, Vyas A, Shi B, et al. Voicebox: Text-guided multilingual universal speech generation at scale[J]. Advances in neural information processing systems, 2024, 36.
>     [r2] Polyak A, Zohar A, Brown A, et al. Movie gen: A cast of media foundation models[J]. arXiv preprint arXiv:2410.13720, 2024.
>
>
> ## Weaknesses
>
> > Insufficient Mathematical Rigor: The paper lacks adequate mathematical depth and rigor. Incorporating more detailed mathematical formulations, proofs, and theoretical underpinnings would strengthen the credibility and scholarly value of the work.
>
> We have provided a formal definition of multi-objective optimization (MOO), detailed the function and significance of Equation (9), and described how evolutionary algorithms (EA) are integrated with flow matching to enhance our method's robustness.
>
>
> > Incorrect Formalization in Line 105: The statement on line 105 is inaccurate. A Multi-Objective Optimization (MOO) problem cannot be "formally" formulated merely as a vector optimization problem. Such a formulation is more of an informal notational convenience rather than a formal definition. Clarification and correct formalism are needed here.
>
> We have revised the manuscript to provide a more formal and rigorous definition of the multi-objective optimization (MOO) problem, addressing the concern about the original informal notation.
>
> > Unclear Purpose of Equation (9): The role and application of Equation (9) are not clearly explained. It is difficult to understand its significance and how it fits into the overall argument of the paper. Providing a thorough explanation or context for this equation would enhance comprehension.
>
> Equation (9) follows standard ODE sampling, with the exception of the added noise term. In our approach, each sample is associated with a weight vector, and we aim to find suitable samples along its sampling path to optimize the weighted objective. Without the noise term, the sampling trajectory becomes deterministic after the initial random noise is set, restricting exploration to a single path. Introducing the noise term through the Euler–Maruyama method enhances diversity and exploration, allowing the sampling process to generate a wider range of samples from each trajectory and achieve better coverage of the objective space. When there are $K$ neighboring samples, this method facilitates exploration of $K$ times the space in a single pass, eliminating the need to run the sampling path $K$ separate times, thus improving efficiency. We have investigated the impact of K in Figure 4, and a moderate increase of K does improve performance.

---

> > ### Author Response · Authors · 2024-11-16
> >
> > > Unconvincing Integration of EA and Flow Matching
> >
> > Our ParetoFlow method combines flow matching and EA. Our guided vector field is essentially a special case of classifier guidance, with the distinction that we use the uniform weight concept from EA. We believe this point is clear, so we did not elaborate further on it.
> >
> > Another crucial point to clarify is the connection between the flow sampling process and EA. In our flow sampling process, each intermediate noisy sample $\boldsymbol{x}_t$ can be mapped to a clean sample $\boldsymbol{x}_1(\boldsymbol{x}_t)$. This mapping bridges flow matching and EA, where flow matching handles $\boldsymbol{x}_t$ and EA operates on $\boldsymbol{x}_1$.
> >
> > Specifically, in our algorithm, we use the weighted predictor $f_w(\boldsymbol{x}_1(\boldsymbol{x}_t))$ to select promising $\boldsymbol{x}_t$ for the next iteration. This predictor selection, originally part of EA and applied to $\boldsymbol{x}_1$, is integrated into flow matching through its application to $\boldsymbol{x}_t$. This relationship demonstrates the integration of EA and flow matching.
> >
> > To illustrate the efficacy of integrating EA with flow matching, consider the performance of each component in isolation. Removing flow matching from our framework leaves us solely with EA. Table 1 demonstrates that the EA method NSGA-II, performs worse than our integrated approach, highlighting the value added by flow matching. Conversely, excluding EA results in a system where flow matching operates on uniformly weighted objectives for guided sampling, but lacks the crucial selection and local filtering processes. Our experiments on tasks like MO-Hopper and C-10/MOP1 show that this configuration leads to inferior hypervolume (HV) results, as depicted below, further validating the significance of combining both strategies:
> >
> > | Methods                                | C-10/MOP1     | MO-Hopper     |
> > | -------------------------------------- | ------------- | ------------- |
> > | ParetoFlow w/o selection and filtering | $4.53 ± 0.00$ | $5.06 ± 0.00$ |
> > | ParetoFlow (ours)                      | $4.77 ± 0.00$ | $5.69 ± 0.03$ |
> >
> > This comparative analysis clearly supports the effectiveness of our integrated method, demonstrating that each component contributes significantly to the overall performance. We have detailed this synergistic relationship and its empirical validation in the Appendix A.12 titled "Relationship Between Evolutionary Algorithms and Flow Matching".
> >
> >
> >
> > > Inappropriateness of Section 4.6 in Main Text: Section 4.6 may not be suitable for inclusion in the main body of the paper. It might be more appropriate to place this section in an appendix or supplementary material to maintain the focus and flow of the main content.
> >
> > We have revised the paper and moved Section 4.6 to Appendix A.3 to ensure the main content remains focused and maintains a coherent flow.
> >
> > > Unconvincing Experimental Results Using Hypervolume Metric: The experiments conducted are not entirely convincing, particularly due to the exclusive use of the hypervolume (HV) metric. If alternative methods such as SMS-MOEA were employed, it is questionable whether the proposed method would perform as well as, or better than, SMS-MOEA. Including comparisons with such methods and using a variety of performance metrics could provide a more comprehensive evaluation.
> >
> > We have expanded our analysis to include widely recognized methods such as NSGD-III and SMS-EMOA, applied to the same five tasks highlighted in our ablation studies. Our findings demonstrate that ParetoFlow consistently outperforms these traditional approaches, reinforcing the effectiveness and robustness of our method. The results are as follows:
> >
> >
> > | Methods           | ZDT2          | C-10/MOP1     | MO-Hopper     | Zinc          | RE23          |
> > | ----------------- | ------------- | ------------- | ------------- | ------------- | ------------- |
> > | NSGD-III + MM     | $5.74 ± 0.05$ | $4.71 ± 0.01$ | $5.31 ± 0.13$ | $4.15 ± 0.06$ | $4.96 ± 0.04$ |
> > | SMS-EMOA + MM     | $6.23 ± 0.09$ | $4.73 ± 0.00$ | $5.45 ± 0.23$ | $4.33 ± 0.09$ | $5.67 ± 0.08$ |
> > | ParetoFlow (ours) | $6.79 ± 0.16$ | $4.77 ± 0.00$ | $5.69 ± 0.03$ | $4.49 ± 0.06$ | $6.32 ± 0.46$ |
> >
> > These results have been included in the Appendix A.2 titled "Extended Comparisons" for detailed review.
> >
> > In addition to the hypervolume (HV) metric, another common measure is the inverted generational distance (IGD), which evaluates the proximity of the solution set to the true Pareto front. However, for our real-world tasks, the true Pareto front is unknown, making the IGD metric inapplicable. This constraint highlights the necessity of using HV for our evaluations and supports our methodological choices.
> >
> > ## Overall
> >
> > Thank you for your thoughtful review. We greatly value your feedback and hope this response meets your expectations. Does it address your concerns? We look forward to further exchange during the rebuttal phase.

---

> > > ### Author Response · Authors · 2024-11-21
> > > **Looking Forward to Your Feedback**
> > >
> > > Dear Reviewer Cej3,
> > >
> > > We sincerely appreciate your thoughtful review and constructive feedback. To address your concerns, we have carefully addressed your concerns as follows:
> > >
> > >
> > > - We have added two additional citations and conducted more experiments on MOO to demonstrate the effectiveness of flow matching over diffusion models.
> > > - We have provided a formal definition of multi-objective optimization.
> > > - We have clarified Equation (9) for improved understanding.
> > > - We have justified and conducted further experiments to demonstrate the effectiveness of integrating EA and flow matching.
> > > - We have revised the paper and moved Section 4.6 to Appendix A.3 to ensure the main content remains focused and maintains a coherent flow.
> > > - We have expanded our analysis to include widely recognized methods such as NSGA-III and SMS-EMOA.
> > >
> > >
> > > Thank you once again for your valuable insights. Please let us know if there are any additional questions or areas requiring further clarification. As the weekend approaches, we welcome the opportunity to continue this discussion. We look forward to your response.

---

> ### Author Response · Authors · 2024-11-26
> **Thank you for your support!**
>
> We are glad to have addressed your concerns. Thank you for taking the time to review our response and for your support in revisiting the evaluation. We truly appreciate your thoughtful consideration and positive feedback.

---

### Official Review · Reviewer_4YDk · 2024-11-03

**Soundness:** 3
**Presentation:** 2
**Contribution:** 3
**Rating:** 6
**Confidence:** 3

**Summary:**

This paper proposes ParetoFlow, a novel method for offline multi-objective optimization (MOO) using flow matching. It introduces a multi-objective predictor guidance module and a neighboring evolution module to improve the efficiency and effectiveness of flow sampling in approximating the Pareto front. The paper presents extensive experimental results on various benchmarks, demonstrating ParetoFlow’s superior performance compared to existing MOO methods.

**Strengths:**

1. The paper is well-written and easy to follow.
2. The proposed modules, multi-objective predictor guidance and neighboring evolution, provide a well-motivated approach to guiding flow sampling and promoting knowledge sharing among neighboring distributions.
3. The experimental results of ParetoFlow on various benchmarks are promising.

**Weaknesses:**

1. The paper mentions the computational cost of ParetoFlow but does not provide a detailed comparison with other generative modeling methods.
2. Providing some visualization results for the generated samples would be beneficial.

**Questions:**

1. Can the authors offer a case study for the generated architectures in terms of structure and performance on MO-NAS?
2. How to guarantee the validity of the generated samples? For instance, an architecture with the valid computational flow in the NAS task.

---

> ### Author Response · Authors · 2024-11-16
>
> Dear Reviewer,
>
> Thank you for your constructive feedback. Your observations have been invaluable for refining our paper. We have made revisions accordingly, with changes marked in red for ease of review.
>
>
> ## Weaknesses
>
> > The paper mentions the computational cost of ParetoFlow but does not provide a detailed comparison with other generative modeling methods.
>
> Thank you for highlighting the need for a detailed computational cost comparison. We have now expanded our analysis to include both generative and traditional methods and incorporated these into Appendix A.5 "Computational Cost". The table below summarizes the computational times (in minutes) for ParetoFlow and other methods across two tasks:
>
> Considering the minimal overhead and significant performance gains of our method compared to baselines, we believe it provides a practical solution for researchers and practitioners seeking effective results without sacrificing speed.
>
>
> | Tasks             | C-10/MOP1 | MO-Hopper |
> | ----------------- | --------- | --------- |
> | E2E               | $1.20$    | $1.17$    |
> | MH                | $1.24$    | $1.17$    |
> | MM                | $1.70$    | $1.80$    |
> | MOEA/D + MM       | $1.50$    | $1.36$    |
> | MOBO              | $0.12$    | $33.68$   |
> | PROUD             | $2.23$    | $4.05$    |
> | LaMBO-2           | $2.54$    | $4.22$    |
> | CorrVAE           | $1.81$    | $2.89$    |
> | MOGFN             | $5.73$    | $10.62$   |
> | ParetoFlow (ours) | $2.29$    | $4.18$    |
>
>
> > Providing some visualization results for the generated samples would be beneficial.
>
> We appreciate the suggestion and have included visualization results for C-10/MOP1 and MO-Hopper in the Appendix A.8 titled "Visualization and Case Study Details." The appendix features comparisons between offline samples and samples generated by ParetoFlow, clearly demonstrating the superior quality of the latter.
>
> ## Questions.
>
> > Can the authors offer a case study for the generated architectures in terms of structure and performance on MO-NAS?
>
> We have incorporated a detailed case study on C-10/MOP5 in our Appendix A.8 titled "Visualization and Case Study Details." This case study focuses on optimizing prediction error, model complexity, and hardware efficiency, as outlined in [r1].
>
> The case study analyzes three sets of solutions where the weight vector for unconsidered objectives is set to zero:
> 1. **Prediction Error and Model Complexity**: This set emphasizes minimizing prediction error and model complexity.
> 2. **Prediction Error and Hardware Efficiency**: This set prioritizes reducing prediction error with improved hardware efficiency.
> 3. **Balanced set**: This set aims to balance prediction error, model complexity, and hardware efficiency.
>
> We discuss the frequency of operators used in these sets, noting that the 3x3 convolution is consistently preferred for its effectiveness in reducing prediction error, while the 3x3 average pooling is less favored. Additionally, there is a notable shift from the use of 1x1 convolutions to 'none' operators in moving from the first to the second set, suggesting a trade-off for better GPU latency.
>
> This analysis provides insights into the structural preferences and performance trade-offs in the generated architectures.
>
>
>      [r1] Lu Z, Cheng R, Jin Y, et al. Neural architecture search as multiobjective optimization benchmarks: Problem formulation and performance assessment[J]. IEEE transactions on evolutionary computation, 2023, 28(2): 323-337.
>
>
> > How to guarantee the validity of the generated samples? For instance, an architecture with the valid computational flow in the NAS task.
>
> To ensure that our generated samples are always valid, we adopt a strategy of encoding discrete architectures into continuous logits, as suggested by [r2, r3]. This approach allows us to train our models effectively in a continuous space. During the sampling process, we generate samples in the continuous (logit) space and then decode them back into the discrete architecture space. The encoding and decoding processes are straightforward and designed to preserve the integrity of the structures, ensuring that the decoded samples are valid architectures.
>
>
>     [r2] Trabucco B, Geng X, Kumar A, et al. Design-bench: Benchmarks for data-driven offline model-based optimization[C]//International Conference on Machine Learning. PMLR, 2022: 21658-21676.
>     [r3] Xue K, Tan R X, Huang X, et al. Offline Multi-Objective Optimization[J]. arXiv preprint arXiv:2406.03722, 2024.
>
> ## Overall
>
> We appreciate your thorough review and valuable feedback. Thank you for your insights. Does this reply address your main concerns? We look forward to more discussion in the rebuttal.

---

> > ### Author Response · Authors · 2024-11-21
> > **Looking Forward to Your Feedback**
> >
> > Dear Reviewer 4YDk,
> >
> > Thank you for your detailed review and constructive feedback. We have carefully addressed your concerns and revised the submission accordingly. Specifically:
> >
> > - We have provided a detailed comparison of ParetoFlow with other generative modeling methods, focusing on computational cost.
> > - We have included visualization results for C-10/MOP1 and MO-Hopper to further demonstrate the effectiveness of ParetoFlow.
> > - We have incorporated a detailed case study on C-10/MOP5.
> > - We have clarified how we ensure the validity of the generated samples.
> >
> >
> > We sincerely appreciate your valuable feedback and are happy to address any further questions or concerns. As the weekend approaches, we are eager to engage with you further. Thank you again, and we look forward to your response.

---

> > ### Comment · Reviewer_4YDk · 2024-11-26
> > **Thanks for your detailed response**
> >
> > Thanks for your detailed response. Most of my concerns have been addressed. I will keep my positive rating. I advise to visualize some generated architectures with good performance at the instance level for better understanding.

---

> > > ### Author Response · Authors · 2024-11-26
> > > **Thank you for your support!**
> > >
> > > We are pleased to have addressed most of your concerns. We appreciate your positive feedback and suggestion to visualize the generated architectures, which we believe will enhance the clarity of our method.  We will include these visualizations in the final version of the paper.

---

### Official Review · Reviewer_YcyJ · 2024-11-04

**Soundness:** 3
**Presentation:** 3
**Contribution:** 3
**Rating:** 6
**Confidence:** 4

**Summary:**

This paper presents *ParetoFlow*, an innovative approach aimed at offline multi-objective optimization (MOO) that leverages the flow-matching generative modeling framework. The authors introduce two key modules: a *multi-objective predictor guidance module* that uses uniform weight vectors to direct sample generation toward approximating the Pareto front and a *neighboring evolution module* for sharing knowledge between distributions with similar objectives. These techniques are designed to improve coverage of both convex and non-convex Pareto fronts and enhance sample diversity.

**Strengths:**

- The integration of flow matching in offline MOO is a fresh perspective compared to the typical use of evolutionary algorithms or Bayesian optimization. This method captures the potential of generative models for addressing complex, multi-faceted optimization challenges.

- The proposed module that fosters information exchange among adjacent distributions is a standout feature. This knowledge-sharing mechanism supports enhanced exploration and refinement of solutions.

- This method was evaluated across diverse benchmarks, demonstrating state-of-the-art performance and outperforming existing MOO methods such as PROUD, LaMBO-2, and CorrVAE, especially in generating high-quality Pareto-optimal solutions.

- The provided code and training details are clear.

**Weaknesses:**

- You only cited [1] to state that Flow matching outperforms Diffusion models, which I find insufficient. This is especially true in the context of multi-objective optimization, where specific problems require specialized analysis.

- Although the results on benchmarks with high-dimensional objectives, such as MO-NAS, are mentioned, there could be more detailed insights into how well the method scales with increased complexity in terms of computational cost and solution quality.

- The comparisons focus more on generative modeling approaches. A broader comparison that includes more traditional, non-generative MOO techniques could strengthen the context of ParetoFlow's advantages and limitations.

[1] Yaron Lipman, Ricky TQ Chen, Heli Ben-Hamu, Maximilian Nickel, and Matt Le. Flow matching for generative modeling.

**Questions:**

1. How does ParetoFlow manage potential overfitting to specific tasks or datasets, given that it optimizes multiple objectives using a learned generative model?

2. Can the authors elaborate on the choice of the Das-Dennis method for generating uniform weights? Are there scenarios where alternative weight distribution methods could be more effective?

I am open to revising my score upwards if the authors can satisfactorily address these concerns during the discussion phase.

---

> ### Author Response · Authors · 2024-11-16
>
> Dear Reviewer,
>
> We greatly appreciate your insightful feedback, which has played a crucial role in improving our manuscript. We have thoughtfully addressed your suggestions and highlighted all revisions in red to facilitate your review.
>
>
> ## Weaknesses
>
> > You only cited [1] to state that Flow matching outperforms Diffusion models, which I find insufficient. This is especially true in the context of multi-objective optimization, where specific problems require specialized analysis.
>
>
> We have now included two additional high-impact papers [r1, r2] in our revised submission to provide stronger evidence that flow matching outperforms diffusion models. Specifically, Section 5.5 (Ablation on generative modeling approaches) in [r1] and Section 3.6.2 (Ablations) in [r2] address this topic in detail.
>
> Furthermore, to empirically substantiate our method's efficacy, we implemented a variant of our ParetoFlow framework using a diffusion model and conducted comparative experiments on two tasks, MO-Hopper and C-10/MOP1, using five random seeds. The results, detailed in the table below and added to our Appendix A.10 titled "Further Ablations", clearly demonstrate the superior performance of flow matching:
>
>
> | Methods                 | C-10/MOP1     | MO-Hopper     |
> | ----------------------- | ------------- | ------------- |
> | ParetoFlow w/ Diffusion | $4.65 ± 0.05$ | $5.54 ± 0.07$ |
> | ParetoFlow (ours)       | $4.77 ± 0.00$ | $5.69 ± 0.03$ |
>
> These empirical findings corroborate our choice of flow matching over diffusion models, supporting our claim with both qualitative and quantitative evidence.
>
>      [r1] Le M, Vyas A, Shi B, et al. Voicebox: Text-guided multilingual universal speech generation at scale[J]. Advances in neural information processing systems, 2024, 36.
>      [r2] Polyak A, Zohar A, Brown A, et al. Movie gen: A cast of media foundation models[J]. arXiv preprint arXiv:2410.13720, 2024.
>
>
>
> >  detailed insights into how well the method scales
>
>
> To provide more detailed insights, we have conducted a thorough analysis across a diverse set of MO-NAS tasks from the NAS-Bench series, including C-10/MOP1, MOP2, MOP5, MOP6, and MOP7.
>
>
> | Tasks                     | C-10/MOP1 | C-10/MOP2 | C-10/MOP5 | C-10/MOP6 | C-10/MOP7 | MO-Hopper | MO-Swimmer |
> | ------------------------- | --------- | --------- | --------- | --------- | --------- | --------- | ---------- |
> | raw input dimension       | $26$      | $26$      | $6$       | $6$       | $6$       | $10184$   | $9734$     |
> | input dimension (logits)  | $31$      | $31$      | $24$      | $24$      | $24$      | $N/A$     | $N/A$      |
> | number of objectives      | $2$       | $3$       | $5$       | $6$       | $8$       | $2$       | $2$        |
> | number of offline samples | $12084$     | $26316$     | $9375$      | $9375$      | $9375$      | $4500$      | $8571$       |
> | Predictor training (min)  | $0.76$    | $2.44$    | $1.45$    | $1.80$    | $2.38$    | $1.03$    | $1.84$     |
> | Flow model training (min) | $1.28$    | $5.92$    | $1.20$    | $5.18$    | $1.19$    | $2.51$    | $2.53$     |
> | Design sampling (min)     | $0.25$    | $0.52$    | $0.41$    | $0.44$    | $0.51$    | $0.64$    | $0.60$     |
> | Overall time cost (min)   | $2.29$    | $8.88$    | $3.06$    | $7.42$    | $4.08$    | $4.18$    | $4.97$     |
> | Rank of ParetoFlow        | $1$       | $1$       | $1$       | $1$       | $1$       | $1$       | $1$        |
>
>
> Our findings indicate that despite variations in the number of objectives and design dimensions, the computational cost of our method remains consistent. Notably, for tasks with higher dimensional objectives such as MO-Swimmer and MO-Hopper, our computational efficiency is comparable to tasks with lower dimensions. This consistency underscores our method's scalability across a range of computational demands. Additionally, our method consistently achieves strong performance, further attesting to its robustness. We have detailed these results in the table provided above, which compares key metrics across tasks, and included more in-depth discussion in the Appendix A.5 titled "Computational Cost".
>
>
> In terms of solution quality, we adopt a strategy of encoding discrete architectures into continuous logits, as suggested by [r3, r4]. This approach allows us to train our models effectively in a continuous space. During the sampling process, we generate samples in the continuous (logit) space and then decode them back into the discrete architecture space.  This encoding and decoding process is straightforward and ensures that the structural integrity of the architectures is maintained, resulting in valid and high-quality decoded samples.
>
>     [r3] Trabucco B, Geng X, Kumar A, et al. Design-bench: Benchmarks for data-driven offline model-based optimization[C]/ ICML 2022
>     [r4] Xue K, Tan R X, Huang X, et al. Offline Multi-Objective Optimization[J]. ICML 2024

---

> > ### Author Response · Authors · 2024-11-16
> >
> > > The comparisons focus more on generative modeling approaches. A broader comparison that includes more traditional, non-generative MOO techniques could strengthen the context of ParetoFlow's advantages and limitations.
> >
> >
> > We have expanded our analysis to include widely recognized methods such as NSGD-III and SMS-EMOA [r5], applied to the same five tasks highlighted in our ablation studies. Our findings demonstrate that ParetoFlow consistently outperforms these traditional approaches, reinforcing the effectiveness and robustness of our method. The results are as follows:
> >
> >
> >
> > | Methods           | ZDT2          | C-10/MOP1     | MO-Hopper     | Zinc          | RE23          |
> > | ----------------- | ------------- | ------------- | ------------- | ------------- | ------------- |
> > | NSGD-III + MM     | $5.74 ± 0.05$ | $4.71 ± 0.01$ | $5.31 ± 0.13$ | $4.15 ± 0.06$ | $4.96 ± 0.04$ |
> > | SMS-EMOA + MM     | $6.23 ± 0.09$ | $4.73 ± 0.00$ | $5.45 ± 0.23$ | $4.33 ± 0.09$ | $5.67 ± 0.08$ |
> > | ParetoFlow (ours) | $6.79 ± 0.16$ | $4.77 ± 0.00$ | $5.69 ± 0.03$ | $4.49 ± 0.06$ | $6.32 ± 0.46$ |
> >
> > These results have been included in the Appendix A.2 titled "Extended Comparisons" for detailed review.
> >
> >     [r5] Beume N, Naujoks B, Emmerich M. SMS-EMOA: Multiobjective selection based on dominated hypervolume[J]. European Journal of Operational Research, 2007, 181(3): 1653-1669.
> >
> >
> >
> > ## Questions:
> >
> >
> > > How does ParetoFlow manage potential overfitting to specific tasks or datasets, given that it optimizes multiple objectives using a learned generative model?
> >
> >
> > ParetoFlow independently trains a single flow matching model and multiple property predictors for each task. During training, we use a validation set to select the best checkpoint for both the flow matching model and the predictors, helping to mitigate overfitting risks. Additionally, the flow matching model is designed with built-in resistance to overfitting. Specifically, it employs linear interpolation between real samples and pure noise during training to predict the vector field. This interpolation serves as a robust data augmentation strategy, effectively regularizing the model and reducing the likelihood of overfitting. These measures collectively enhance the generalization capability of ParetoFlow.
> >
> > > Can the authors elaborate on the choice of the Das-Dennis method for generating uniform weights? Are there scenarios where alternative weight distribution methods could be more effective?
> >
> > The Das-Dennis method is widely used in multi-objective optimization studies due to its simplicity and ease of use, as it does not require optimization. However, a limitation of this method is that it does not allow for specifying an exact number of weights. On the other hand, the Riesz s-Energy method [r6] allows for precise control over the number of weights generated. However, this approach involves a optimization process, making it more complex to implement.
> >
> > We conducted experiments on two tasks using both strategies, and found that their performance was close, as shown in the table below. We have included these discussions in Appendix A.10 titled "Further Ablations".
> >
> >
> > | Methods                      | C-10/MOP1     | MO-Hopper     |
> > | ---------------------------- | ------------- | ------------- |
> > | ParetoFlow w/ Riesz s-Energy | $4.77 ± 0.00$ | $5.55 ± 0.10$ |
> > | ParetoFlow (ours)            | $4.77 ± 0.00$ | $5.69 ± 0.03$ |
> >
> >
> >      [r6] Hardin D P, Saff E B. Minimal Riesz energy point configurations for rectifiable d-dimensional manifolds[J]. Advances in Mathematics, 2005, 193(1): 174-204.
> >
> > ## Overall
> >
> > Thank you for your detailed review and feedback. We appreciate your insights and look forward to engaging further during the rebuttal phase. Does our response address your concerns?

---

> > > ### Author Response · Authors · 2024-11-21
> > > **Looking Forward to Your Feedback**
> > >
> > > Dear Reviewer YcyJ,
> > >
> > > Thank you for your detailed review and constructive feedback. We have carefully addressed the concerns you raised and revised the submission accordingly, including:
> > >
> > > - We have added two additional citations and conducted more experiments on MOO to demonstrate the effectiveness of flow matching over diffusion models.
> > > - We have provided detailed insights to validate the scalability of our ParetoFlow method.
> > > - We have included two additional traditional methods, NSGA-III and SMS-EMOA, to further highlight the advantages of ParetoFlow.
> > > - We have discussed how ParetoFlow mitigates potential overfitting.
> > > - We have justified the choice of the Das-Dennis method and compared it with the Riesz s-Energy method.
> > >
> > > We greatly appreciate your valuable feedback and are happy to address any remaining questions or concerns. As the weekend approaches, we would like to ensure we have the opportunity to engage with you further. Thank you again, and we look forward to your response.

---

> > > > ### Comment · Reviewer_YcyJ · 2024-11-26
> > > >
> > > > Thank you for your response. This is an attractive setting for Multi-Objective Optimization, and I appreciate how the authors have addressed my concerns. I would like to keep my positive score.

---

> > > > > ### Author Response · Authors · 2024-11-26
> > > > > **We appreciate your continued support!**
> > > > >
> > > > > Thank you for your thoughtful feedback and for confirming that we have satisfactorily addressed your concerns during the discussion phase. We are pleased that you find our setting for multi-objective optimization attractive.
> > > > >
> > > > > As you mentioned in your initial review that you are open to revising your score upwards, we greatly appreciate your willingness to reconsider our evaluation and are thankful for your continued support and consideration.

---

### Meta-Review · Area_Chair_A8oc · 2024-12-19

**Metareview:**

Summary:
This paper presents ParetoFlow, a method based on the flow-matching generative modeling framework. It integrates two key modules: (1) Multi-Objective Predictor Guidance Module – Utilizes uniform weight vectors to guide sample generation and facilitate approximation of the Pareto front. (2) Neighboring Evolution Module – Encourages knowledge sharing between distributions with similar objectives to enhance the optimization process. These modules collectively aim to improve the coverage of both convex and non-convex Pareto fronts and increase sample diversity.

Strengths:
The concept behind ParetoFlow looks innovative and compelling. Additionally, the experimental results demonstrate promising performance on the tested problems.

Weaknesses:
Most of the concerns raised by the reviewers appear to have been adequately addressed.

Decision:
I recommend acceptance, as the paper is well-executed overall, and the authors have addressed the majority of the reviewers' feedback. However, the authors should carefully incorporate all reviewer suggestions in the final version, such as visualizing the generated architectures, as recommended by one reviewer.

**Additional Comments On Reviewer Discussion:**

All the four reviewers were engaged in discussion with the authors. While the reviewers raised many concerns, most of them have been addressed during rebuttal, which were also acknowledged by the reviewers.

I appreciate the efforts made by the authors during rebuttal, and concur with the reviewers that this paper could be accepted.

---

### Decision · Program_Chairs · 2025-01-22

Accept (Poster)